# Asymmetric retinal direction tuning predicts optokinetic eye movements across stimulus conditions

**Scott C Harris[1,2]\*, Felice A Dunn[1]\***

[1]Department of Ophthalmology, University of California, San Francisco, San Francisco, United States; [2]Neuroscience Graduate Program, University of California, San Francisco, San Francisco, United States

**Abstract** Across species, the optokinetic reflex (OKR) stabilizes vision during self-motion. OKR occurs when ON direction-selective retinal ganglion cells (oDSGCs) detect slow, global image motion on the retina. How oDSGC activity is integrated centrally to generate behavior remains unknown. Here, we discover mechanisms that contribute to motion encoding in vertically tuned oDSGCs and leverage these findings to empirically define signal transformation between retinal output and vertical OKR behavior. We demonstrate that motion encoding in vertically tuned oDSGCs is contrast-sensitive and asymmetric for oDSGC types that prefer opposite directions. These phenomena arise from the interplay between spike threshold nonlinearities and differences in synaptic input weights, including shifts in the balance of excitation and inhibition. In behaving mice, these neurophysiological observations, along with a central subtraction of oDSGC outputs, accurately predict the trajectories of vertical OKR across stimulus conditions. Thus, asymmetric tuning across competing sensory channels can critically shape behavior.

\*For correspondence:
Scott.Harris@ucsf.edu (SCH);
Felice.Dunn@ucsf.edu (FAD)

**Competing interest:** The authors declare that no competing interests exist.

## Editor's evaluation

The optokinetic reflex (OKR) is a behavioral response that moves the eye so as to keep the image on the retina stabilized during the animal's movements. This important study traces the origins of that behavior down to cellular mechanisms of circuits in the retina. Specifically, it offers compelling evidence that the asymmetries and tuning properties of the OKR are shaped by the excitation/inhibition balance in retinal circuits and the regulation of spiking thresholds across different neuron types.

## Introduction

From humans to insects, a wide range of organisms depend on vision to navigate their environments. When these animals respond to incoming visual information by enacting motor plans, however, relative motion is created between the eye and the visual scene. Such motion, termed 'retinal slip,' has the potential to corrupt subsequent vision and threaten survival. To compensate for this possibility, the optokinetic reflex (OKR) is a highly conserved visual behavior that stabilizes retinal image motion across species (including invertebrates [*Zeil et al., 1989*], reptiles, amphibians, fish, birds, and all mammals, reviewed by *Masseck and Hoffmann, 2009*). OKR consists of visually-evoked, compensatory eye movements (or head movements in some species) that offset the slow, global image motion generated by self-movement (*Simpson, 1984*). In most species, OKR also changes across stimulus conditions: adjustments to stimulus contrast, color, location, velocity, or spatial frequency, for example, can elicit distinct OKR patterns (*Collewijn, 1969*; *Donaghy, 1980*; *Gravot et al., 2017*; *Knorr et al.,*

*2021*; *Leguire et al., 1991*; *Cahill and Nathans, 2008*; *Dehmelt et al., 2021*; *Shimizu et al., 2010*). Nonetheless, neurophysiological mechanisms for such phenomena remain unknown.

The anatomical pathways that underlie OKR in mammals are well defined (reviewed by *Masseck and Hoffmann, 2009*, *Simpson, 1984*, *Dhande et al., 2015*, *Giolli et al., 2006*). Starting in the retina, a dedicated class of biological motion detectors known as ON direction-selective retinal ganglion cells (oDSGCs) encode the slow global image motion that occurs during retinal slip. Classic work identified three types of oDSGCs that each spike maximally in response to a different cardinal direction of stimulus motion (i.e., upward/superior, downward/inferior, and nasal/anterior) (*Oyster, 1968*; *Oyster and Barlow, 1967*). A recent study also identified a fourth oDSGC type in mice that encodes temporal/posterior motion (*Sabbah et al., 2017*). Projections from oDSGCs avoid typical retinorecipient structures such as the superior colliculus and lateral geniculate nucleus. Instead, axons from vertically and horizontally preferring oDSGCs course along dedicated retinofugal tracts to a set of midbrain nuclei known collectively as the accessory optic system. Vertically tuned oDSGCs terminate exclusively in, and comprise the sole retinal inputs to, the medial and lateral terminal nuclei (MTN and LTN) (*Dhande et al., 2015*; *Dhande et al., 2013*; *Lilley et al., 2019*; *Sun et al., 2015*; *Yonehara et al., 2008*; *Yonehara et al., 2009*; *van der Togt et al., 1993*) (but see *Kay et al., 2011* and 'Discussion') - though in mice, LTN is engulfed within MTN (*Dhande et al., 2013*; *Yonehara et al., 2009*; *Osterhout et al., 2015*; *Pak et al., 1987*). Here, their inputs are likely integrated into a single velocity signal that reflects the vertical component of retinal slip. This information is then relayed deeper into the brainstem where corresponding eye movements are enacted. Likewise, horizontally tuned oDSGCs exclusively target the nucleus of the optic tract/dorsal terminal nucleus (NOT/DTN) where their signals are similarly integrated to ultimately generate the horizontal component of OKR (*Dhande et al., 2013*; *Osterhout et al., 2015*). While correlations have been recognized between the physiological properties of oDSGCs and OKR – particularly a matched speed tuning (*Oyster et al., 1972*) – little is known about how signals from multiple oDSGC types are integrated to generate OKR across varying stimulus conditions (*Dhande et al., 2015*; *Wei, 2018*).

Here, we reveal a general mechanism by which OKR may be generated across a range of stimulus statistics. First, we focus on a single parameter that is known to affect OKR: motion direction. In many species, including both humans (*Takahashi et al., 1978*; *Hainline et al., 1984*; *Murasugi and Howard, 1989*; *van den Berg and Collewijn, 1988*) (but see *Knapp et al., 2013*) and mice (*Yonehara et al., 2009*), OKR is more robust in response to superior motion than inferior motion. We aim to illuminate the mechanism underlying this asymmetry in order to reveal more general processes by which oDSGC signals are centrally integrated to produce OKR. Focusing on the directional asymmetry of vertical OKR is methodologically strategic in that (1) it limits the source of possible neurophysiological mechanisms to functions of only the vertical OKR pathway, and (2) unlike horizontal OKR, which relies on interhemispheric communication and mirror-image motion signals from each eye (*Masseck and Hoffmann, 2009*; *Dhande et al., 2013*; *Hoffmann and Fischer, 2001*), vertical OKR can be studied unilaterally. Our results indicate that the behavioral asymmetry between superior and inferior OKR is linked to differences in the direction tuning properties of oDSGCs that prefer superior and inferior motion. These physiological differences arise from a shift in the balance of excitatory and inhibitory (E/I) synaptic inputs across cell types, along with nonlinear transformations associated with spike thresholding. More generally, we demonstrate that motion encoding in vertically tuned oDSGCs is uniquely sensitive to such changes in synaptic input weights and show how this sensitivity, along with a central subtraction of oDSGC activity, can account for changes to OKR across additional stimulus conditions in behaving mice.

## Results
### OKR is more robust in the superior than inferior direction
Across species, superior motion generates a more robust OKR than inferior motion (e.g., cat [*Hoffmann and Fischer, 2001*; *Evinger and Fuchs, 1978*; *Grasse and Cynader, 1988*], chicken [*Wallman and Velez, 1985*], monkey [*Takahashi and Igarashi, 1977*], human [*Takahashi et al., 1978*; *Hainline et al., 1984*; *Murasugi and Howard, 1989*; *van den Berg and Collewijn, 1988*]). In mice, this phenomenon has been reported in juvenile animals (*Yonehara et al., 2009*). To investigate whether an asymmetry between superior and inferior OKR exists in adult mice, we designed a behavioral rig to

accurately evoke and quantify vertical OKR in head-fixed animals (*Figure 1A and B*, *Figure 1—figure supplement 2*, 'Materials and methods') (*Denman et al., 2017*). Eye movements were measured in response to vertically drifting, full-field gratings used previously to evoke OKR (*Sun et al., 2015*; *Yonehara et al., 2009*; *Osterhout et al., 2015*; *Yonehara et al., 2016*). Across all adult mice, superior- and inferior-drifting gratings generated distinct, but reproducible eye movements: while superior gratings elicited repetitive slow, upward-drifting eye movements ('slow nystagmus') interleaved with frequent resetting saccades in the opposite direction ('fast nystagmus') (*Figure 1C*), inferior gratings tended to reliably drive only an initial slow nystagmus immediately following stimulus onset, after which fast nystagmuses were infrequent and eye position changed minimally (*Figure 1D*). To quantify these differences, we isolated periods of slow and fast nystagmus post hoc by extracting saccadic eye movements from the raw eye position trace. Superior stimuli elicited more frequent fast nystagmuses than did inferior stimuli (*Figure 1E*). In addition, the total distance traveled during slow nystagmus was greater for superior stimuli (*Figure 1F*). These results demonstrate that vertical OKR is asymmetric in adult mice.

Despite the stark asymmetry between superior and inferior OKR in response to unidirectional drifting gratings, quantifying OKR gain (ratio of eye velocity to stimulus velocity) under these conditions presented challenges due to variability in the OKR waveform across stimulus directions (*Figure 1C and D*). To better quantify gain, we designed a second stimulus in which a grating oscillated sinusoidally between superior and inferior motion while retaining a constant average position. This oscillating grating evoked sequential superior and inferior slow nystagmuses that were phase-locked to the stimulus (*Figure 1G*, *Figure 1—figure supplement 1*). Moreover, gain tended to be higher during the superior stage compared to the inferior stage of individual oscillations (*Figure 1H*). This bias was reflected by an average offset of vertical eye position in the superior direction over the course of a single stimulus oscillation (*Figure 1I*). Taken together, these results demonstrate that superior motion drives a more robust OKR than inferior motion.

## Superior and Inferior oDSGCs have distinct direction tuning properties

While behavioral asymmetries between superior and inferior OKR could arise anywhere along the vertical OKR pathway, a plausible neurophysiological substrate is at the level of the retina where the pathways that encode superior and inferior motion remain distinct. Further, because the ganglion cells that encode superior motion ('Superior oDSGCs') and inferior motion ('Inferior oDSGCs') together serve as an information bottleneck for the remainder of the pathway, any physiological asymmetry between these cell types will propagate to behavior unless specifically corrected for by subsequent circuitry (see 'Discussion'). Thus, we hypothesized that the behavioral asymmetry between superior and inferior OKR may result from physiological differences between Superior and Inferior oDSGCs.

To probe oDSGCs involved in vertical OKR, we made central injections of a fluorescent retrograde tracer into their central target, MTN (*Figure 2A*, *Figure 2—figure supplement 1*). This approach labeled an average of 669 ± 15 retinal ganglion cells (RGCs; N = 20 retinae) across the contralateral retina (*Figure 2B*, *Figure 2—figure supplement 1*). Retrogradely labeled RGCs were then targeted in ex vivo retinae for electrophysiological investigation using epifluorescence, and we independently validated these data by using two-photon targeting in a separate set of experiments (*Figure 2—figure supplements 3 and 5*; epifluorescence targeting was used for all experiments unless otherwise specified in the figure legends). To investigate the direction tuning properties of MTN-projecting RGCs, we made cell-attached recordings from labeled RGCs while presenting a drifting bar stimulus that moved slowly (10°/s) across the retina in eight directions (*Figure 2C*). The parameters of this stimulus matched those of the gratings used to evoke vertical OKR in behaving animals (i.e., equivalent cycle width, wavelength, and speed to the unidirectional gratings). The majority (94.76%) of retrogradely labeled RGCs were direction-selective and preferred either dorsal-to-ventral (n = 116 of 286) (i.e., Superior oDSGCs because these cells detect superior motion in visual space after accounting for inversion of the image by the eye's optics; *Figure 2D*) or ventral-to-dorsal (i.e., Inferior oDSGCs, n = 155 of 286) motion on the retina (*Figure 2E*). In agreement, mosaic analyses of soma locations indicated that retrogradely labeled cells likely consisted of two RGC types (*Figure 2—figure supplement 1*). Both Superior and Inferior oDSGCs invariably had baseline firing rates of approximately 0 Hz (spikes/s: Sup. 0.0134 ± 0.006; Inf. 0.0338 ± 0.016; p=0.13 Sup. vs. Inf.). Nonetheless, Superior oDSGCs tended to produce more total spikes than Inferior oDSGCs in response to the drifting bar stimulus (*Figure 2F*

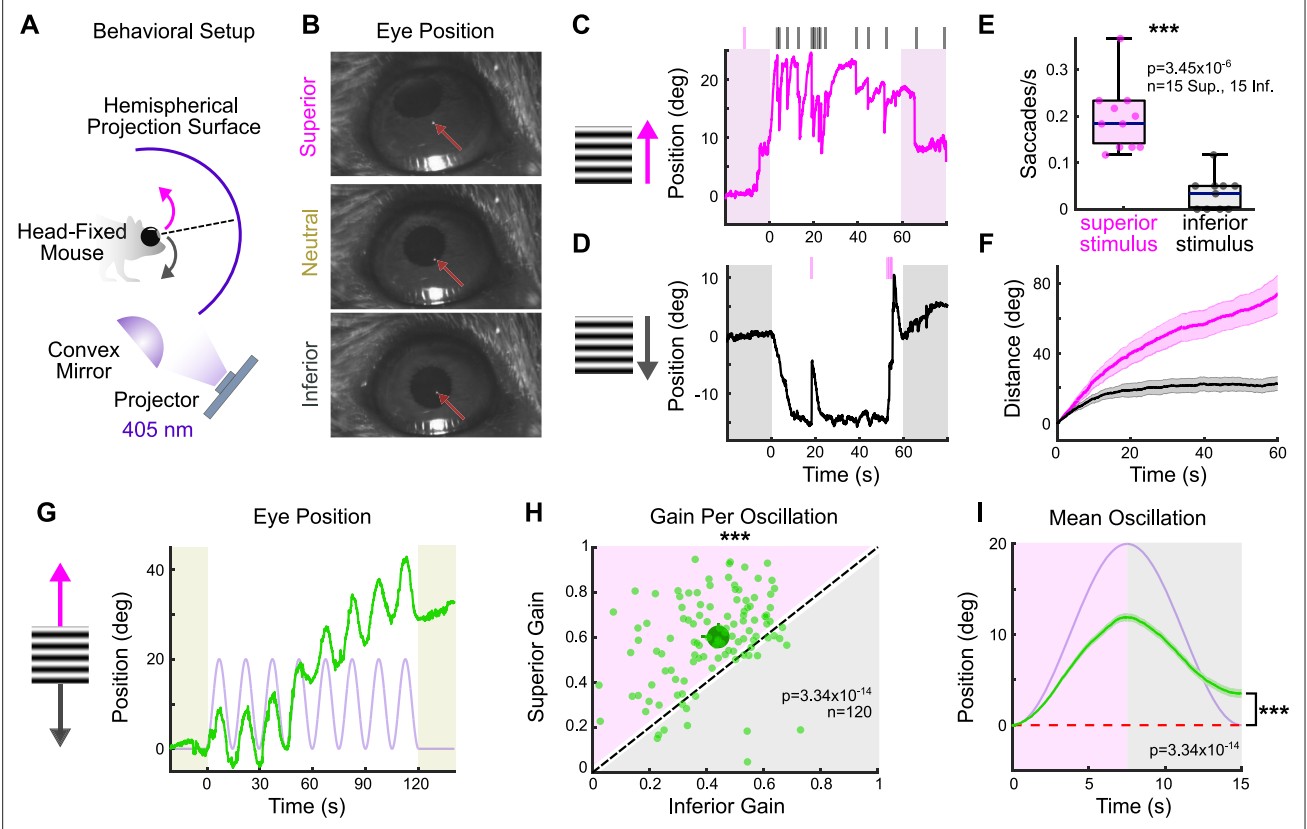

**Figure 1.** The superior and inferior optokinetic reflex (OKR) are asymmetric in adult mice. (**A**) Schematic of behavioral setup to elicit the vertical OKR. The mouse is situated so that one eye is centered in a hemisphere. Stimuli are projected onto the hemisphere's concave surface via reflection off of a convex mirror. Eye movements are tracked using an infrared-sensitive camera and a corneal reflection (see 'Materials and methods'). (**B**) Example video frames demonstrating that the eye traverses between superior, neutral, and inferior positions in the presence of vertically drifting sinusoidal gratings. Red arrows mark the infrared corneal reflection. (**C, D**) Example of OKR in response to full contrast (**C**) superior and (**D**) inferior unidirectional drifting gratings (10°/s). For each epoch, a continuous 60 s stimulus was flanked by 20 s of a static grating (shaded regions). Ticks above the plots mark the time of fast nystagmus either in the superior (magenta) or inferior (gray) direction. Examples from one animal. (**E**) Rate of vertical fast nystagmus for superior and inferior stimuli on each epoch for N = 5 mice. Horizontal line represents median, box boundaries are the interquartile range (IQR), whiskers represent most extreme observation within 1.5× IQR. (**F**) Cumulative vertical distance traveled during slow nystagmus in response to superior (magenta) and inferior (gray) drifting gratings (mean ± SEM). (**G**) Example of OKR in response to a vertically oscillating sinusoidal grating. The eye position is shown in green, and the stimulus position is shown in lavender. Saccades ('fast nystagmuses') have been removed to reveal the asymmetry between superior and inferior OKR. For each epoch, animals viewed eight oscillation cycles lasting a total of 120 s, flanked by 20 s of a static grating (shaded regions). (**H**) Average gain of slow nystagmus during the superior versus inferior stage of individual oscillations. Each small dot is a single oscillation. The region of magenta (or gray) indicates that gain was greater for the superior (or inferior) stage of the oscillation. Points that fall on the line indicate equivalent gain for both stimulus directions. Large dot and whiskers represent univariate medians and 95% confidence intervals (via bootstrapping), respectively. Significance value indicates whether the points tend to fall unevenly on one side of the unity line (two-sided signed-rank). (**I**) Eye position (green) and stimulus position (lavender) averaged across all oscillations and all animals (mean ± SEM). Starting eye position is normalized to 0° at cycle onset. The average ending eye position is displaced in the superior direction (two-sided signed-rank). N = 5 mice for all experiments; n = number of trials. *p<0.05, **p<0.01, ***p<0.001.

The online version of this article includes the following figure supplement(s) for figure 1:

**Figure supplement 1.** Example of sinusoidal vertical optokinetic reflex (OKR) before saccade removal.

**Figure supplement 2.** Baseline vertical eye movements in head-fixed mice (see also *Figure 8—figure supplement 4*).

*and G*), and the total areas of their tuning curves were greater (*Figure 2H*, *Figure 2—figure supplement 3A*). Similarly, we noticed topographic differences within cell types, with MTN-projecting RGCs in dorsal retina spiking more than those in ventral retina (*Figure 2—figure supplement 5*). However, the firing rates of Superior and Inferior oDSGCs covaried, with Superior oDSGCs spiking more than Inferior oDSGCs in every retinal quadrant (*Figure 2—figure supplement 4*).

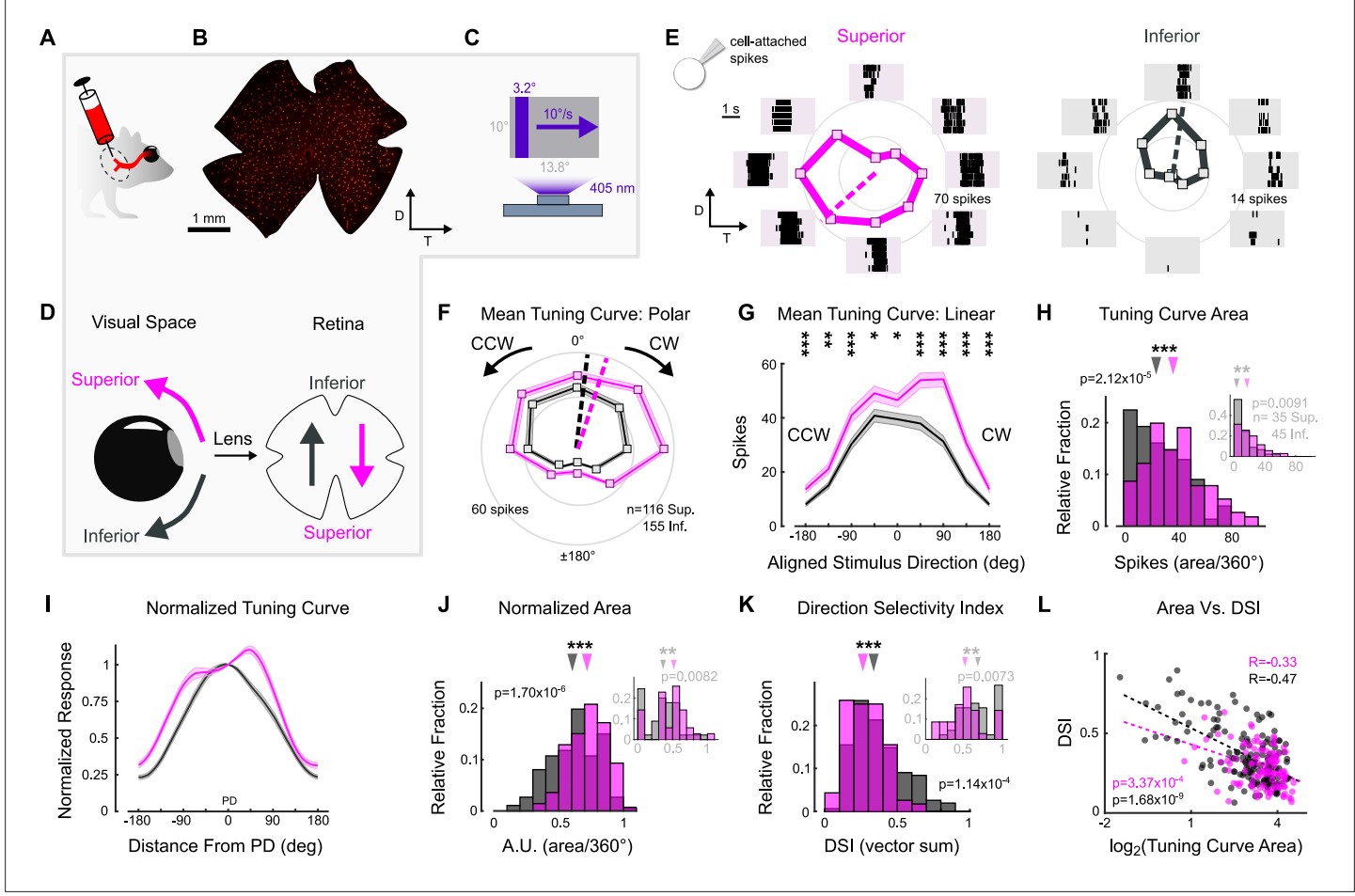

**Figure 2.** Superior and Inferior ON direction-selective retinal ganglion cells (oDSGCs) have asymmetric spike tuning curves. (**A**) Schematic illustrating unilateral bead injections into medial terminal nucleus (MTN) to retrogradely label ganglion cells in the contralateral retina. (**B**) Flat-mount retina with retrogradely labeled, MTN-projecting retinal ganglion cells. (**C**) Drifting bar stimulus (3.2° × limiting projector dimension, 10°/s, 2.4 × 10⁴ S-cone photoisomerization/s). (**D**) Definitions of superior (magenta) and inferior (gray) motion in visual space and on the retina. Directions are inverted by the lens. (**E**) Cell-attached spikes from labeled, MTN-projecting retinal ganglion cells in a flat-mount retina in response to a bar drifting in eight directions. Spike responses and average tuning curves from example Superior (left, magenta) and Inferior (right, gray) oDSGCs. Mean spike counts are presented as the distance from the origin, marked by concentric circles. Numbers on circles indicate spike counts. Dashed lines represent the preferred direction of each cell, calculated as the direction of the vector sum of all responses. Coordinates are in retinal space. (**F, G**) Population tuning curves across all Superior and Inferior oDSGCs (mean ± SEM). (**F**) Polar plots (as in [**E**]) aligned by rotating the tuning curves of Superior cells by 180°. (**G**) Linear representation of the same data (referred to as the 'linear tuning curve'). CW: clockwise, nasal for Superior oDSGCs, temporal for Inferior oDSGCs; CCW: counterclockwise, temporal for Superior oDSGCs, nasal for Inferior oDSGCs. 0° represents directly superior/inferior motion. (**H**) Histograms of the area under the curve of the linear tuning curve of every cell. Inset shows the same metric for a stimulus at 20% relative contrast. (**I**) Population mean (± SEM) normalized tuning curves – computed by normalizing and aligning (at 0°) the response of each cell to its response in the preferred direction. (**J**) Histograms of the area under the curve of the normalized tuning curve (as in [**I**]) of every cell (referred to as 'normalized area'). A larger normalized area indicates a wider tuning curve. Inset shows the same metric for a stimulus at 20% relative contrast. (**K**) Histograms of the direction selectivity index (DSI, vector sum, see 'Materials and methods') of every cell. Inset shows the same metric for a stimulus at 20% relative contrast. (**L**) Linear tuning curve area (as in [**H**]) and direction selectivity index (as in [**K**]) were correlated on a cell-by-cell basis for both Superior and Inferior oDSGCs. Dashed lines are least squares linear regressions, R and p values are Spearman's rank correlation coefficient and associated two-sided significance value, respectively. For all histograms, medians of Superior (magenta) and Inferior (gray) oDSGC distributions are indicated by arrows. *p<0.05, **p<0.01, ***p<0.001.

The online version of this article includes the following figure supplement(s) for figure 2:

**Figure supplement 1.** Two retinal ganglion cell types project to the medial terminal nucleus.

**Figure supplement 2.** Additional metrics of ON direction-selective retinal ganglion cell (oDSGC) spike tuning curve width.

**Figure supplement 3.** Asymmetries between Superior and Inferior ON direction-selective retinal ganglion cells (oDSGCs) persist under two-photon targeting.

*Figure 2 continued on next page*

*Figure 2 continued*

**Figure supplement 4.** Physiological differences between Superior and Inferior ON direction-selective retinal ganglion cells (oDSGCs) are consistent across retinal topography.

**Figure supplement 5.** Topographic variation in direction tuning properties across the retina revealed by two-photon targeting.

We wondered whether the difference in response magnitude between Superior and Inferior oDSGCs could be explained by a simple scaling difference of their tuning curves (i.e., same shape, different size) or whether it was instead associated with an asymmetry in tuning curve shape. To answer this question, we normalized and aligned the tuning curve of each cell in our dataset to its preferred direction (vector sum, see 'Materials and methods') response (*Figure 2I*). If the tuning curves of Superior oDSGCs were scaled versions of those of Inferior oDSGCs, then this normalization and alignment procedure would eliminate any apparent differences between cell types. Instead, however, we found that the normalized tuning curves of Superior oDSGCs had greater widths at 50% response magnitude (*Figure 2—figure supplement 2A*) and total areas (*Figure 2J*, *Figure 2—figure supplement 3C*), indicating that their tuning curves were broader than those of Inferior oDSGCs. In agreement, circular Gaussian fits (see 'Materials and methods') of Superior oDSGC tuning curves were consistently wider than those of Inferior oDSGCs (*Figure 2—figure supplement 2B*). To understand how this difference in tuning curve width might affect the ability of Superior and Inferior oDSGCs to encode motion, we quantified the direction selectivity index (DSI) of each cell (magnitude of the vector sum divided by the scalar sum, see 'Materials and methods'). In agreement, the DSIs of Superior oDSGCs were lower than those of Inferior oDSGCs (*Figure 2K*, *Figure 2—figure supplement 3B*). Finally, we tested whether these asymmetries persisted across stimulus conditions by using a drifting bar stimulus with fivefold lower contrast. As before, Superior oDSGCs spiked more and had broader tuning curves in response to lower contrast bars (*Figure 2H, J and K* insets).

Together, these data indicate that asymmetries in vertical OKR are concomitant with prominent physiological differences between Superior and Inferior oDSGCs: Superior oDSGCs not only spike more than Inferior oDSGCs, but also have broader tuning curves. Further, tuning curve size (i.e., area of the unnormalized tuning curve) and width (i.e., direction selectivity index) were correlated on a cell-by-cell basis (*Figure 2L*, *Figure 2—figure supplement 3D*), indicating that asymmetries in these metrics could arise from a common mechanism.

## Superior oDSGCs receive more excitatory input than Inferior oDSGCs

We sought to determine the source of tuning curve asymmetries between Superior and Inferior oDSGCs. As in the more widely studied class of direction-selective retinal ganglion cell known as the ON-OFF DSGC (ooDSGC) (*Briggman et al., 2011*; *Fried et al., 2002*; *Wei et al., 2011*; *Ding et al., 2016*), oDSGCs inherit the bulk of their direction selectivity via greater inhibition from starburst amacrine cells (SACs) in response to null direction stimuli (*Amthor et al., 2002*; *Yonehara et al., 2011*; *Yoshida et al., 2001*; reviewed by *Wei, 2018*, *Mauss et al., 2017*, *Vaney et al., 2012*). Therefore, we postulated that the difference in tuning curve width between Superior and Inferior oDSGCs may result from asymmetric inhibitory inputs between the two cell types.

To investigate this possibility, we made whole-cell voltage-clamp recordings at the reversal potential for excitation to isolate inhibitory inputs to Superior and Inferior oDSGCs in response to the drifting bar stimulus (*Figure 3A and B*). Across cells, we found no significant difference in the magnitude of inhibitory postsynaptic currents (IPSCs) between Superior and Inferior oDSGCs (*Figure 3C*, *Figure 3—figure supplement 1A*). IPSCs in Superior oDSGCs, however, were slightly more direction-selective than those in Inferior oDSGCs (*Figure 3D*), which is unlikely to explain their broader spike tuning curves (see 'Discussion'). Thus, the canonical model of retinal direction selectivity involving inhibition cannot account for the differences between Superior and Inferior oDSGC spike tuning curves.

We next asked whether excitatory inputs could better explain the asymmetries between the spike tuning curves of Superior and Inferior oDSGCs. To test this possibility, we made voltage-clamp recordings at the reversal potential for inhibition to isolate excitatory postsynaptic currents (EPSCs) during the drifting bar stimulus (*Figure 3E and F*). Across stimulus directions, EPSCs in Superior oDSGCs were between 1.4 and 2.3 times greater than those in Inferior oDSGCs (*Figure 3G*, *Figure 3—figure supplement 1B*). EPSCs were also less direction-selective in Superior than in Inferior oDSGCs (*Figure 3H*). In agreement, the ratio of the peak EPSC to the peak IPSC (E/I) was greater for Superior

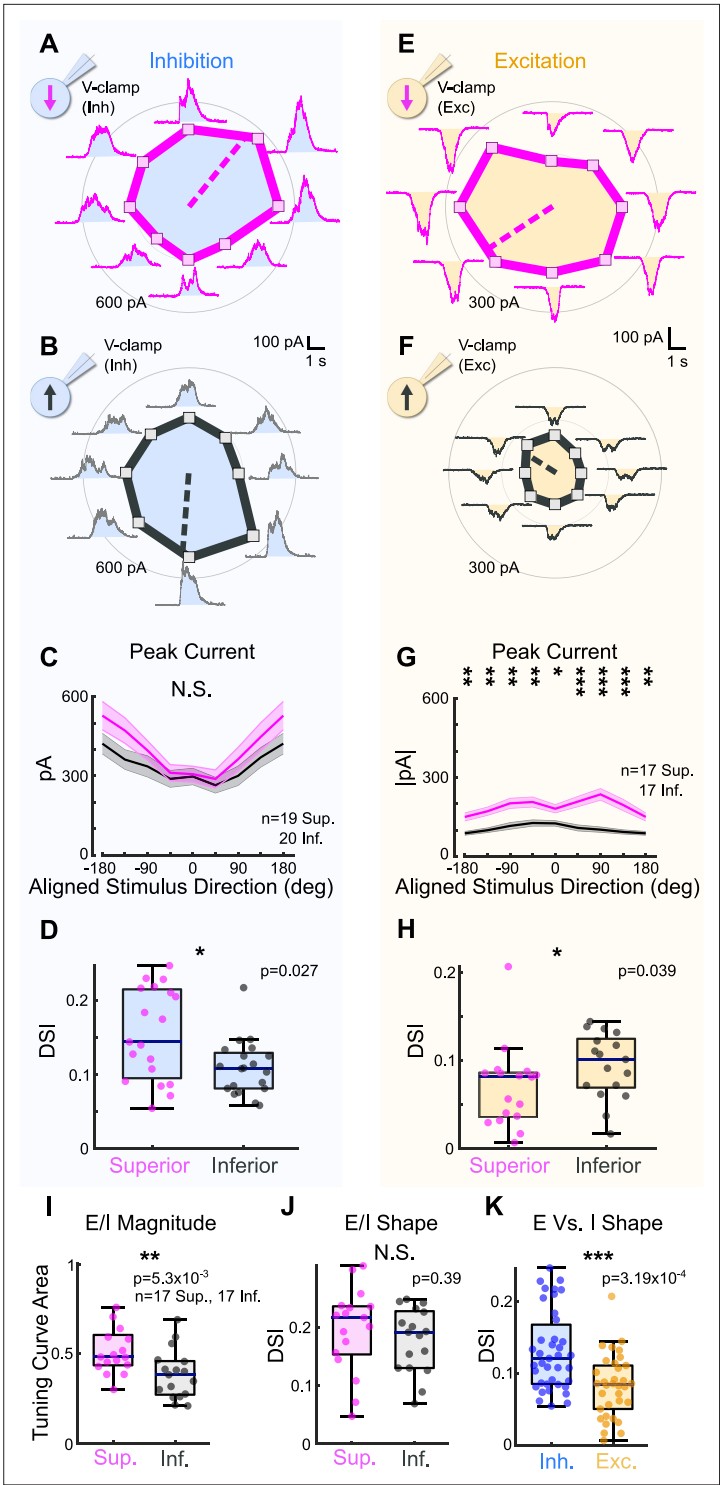

**Figure 3.** Superior ON direction-selective retinal ganglion cells (oDSGCs) receive similar inhibitory inputs but greater excitatory inputs compared to Inferior oDSGCs. (**A**) Inhibitory currents measured from an exemplar Superior oDSGC under voltage-clamp at +10 mV in response to a bar drifting in eight directions. Mean peak inhibitory current is presented as the distance from the origin for each stimulus direction. Dashed line indicates the preferred direction of the peak inhibitory currents. Coordinates are in retinal space. (**B**) Same as (**A**) for an exemplar Inferior oDSGC. (**C**) Population responses for peak inhibitory currents across stimulus directions for Superior (magenta) and Inferior (gray) oDSGCs (mean ± SEM). Stimulus directions are aligned across cell types, where 0° indicates directly superior (for Superior oDSGCs) or inferior (for Inferior oDSGCs) motion. Positive directions are

*Figure 3 continued on next page*

*Figure 3 continued*

clockwise. (**D**) Distributions of the direction selectivity index for peak inhibitory currents in individual Superior and Inferior oDSGCs. (**E**) Excitatory currents measured from an exemplar Superior oDSGC under voltage-clamp at –60 mV in response to a bar drifting in eight directions. Same cell as in (**A**). (**F**) Same as (**E**) for an exemplar Inferior oDSGC. Same cell as in (**B**). (**G**) Population responses for peak excitatory currents across stimulus directions for Superior (magenta) and Inferior (gray) oDSGCs (mean ± SEM). (**H**) Distributions of the direction selectivity index for peak excitatory currents in individual Superior and Inferior oDSGCs. (**I, J**) The ratio of the peak excitatory current to the peak inhibitory current (E/I) was calculated for each stimulus direction for cells in which both metrics were recorded. (**I**) Distributions of the linear tuning curve area of E/I. (**J**) Distributions of the direction selectivity index for E/I. (**K**) Direction selectivity index for peak inhibitory (blue) and excitatory (yellow) currents collapsed across Superior and Inferior oDSGCs. For box plots, horizontal line represents median, box boundaries are IQR, and whiskers represent the most extreme observation within 1.5× IQR. *p<0.05, **p<0.01, ***p<0.001.

The online version of this article includes the following figure supplement(s) for figure 3:

**Figure supplement 1.** Superior ON direction-selective retinal ganglion cells (oDSGCs) receive more excitatory input, but are less intrinsically excitable, than Inferior oDSGCs.

**Figure supplement 2.** Full-field light increments elicit more spikes and excitation in Superior ON direction-selective retinal ganglion cells (oDSGCs).

oDSGCs than for Inferior oDSGCs (*Figure 3I*, *Figure 3—figure supplement 1C*), though not different in direction selectivity (*Figure 3J*). We found no difference in the relative timing of peak EPSCs and peak IPSCs across cell types (not shown). Based on these results, the difference in spike tuning curve size and shape between Superior and Inferior oDSGCs may be related to a corresponding shift in the balance of E/I, associated with an asymmetry in the amount of net excitation that each cell type receives.

To test whether this difference in the magnitude of excitatory input to Superior and Inferior oDSGCs generalized across stimulus types, we measured the spike responses and postsynaptic currents of both cell types in response to a full-field light increment (*Figure 3—figure supplement 2*). As with the drifting bar, the light increment elicited more total spikes in Superior than in Inferior oDSGCs. We also observed significant correlations between the magnitude of a cell's increment response and both the area of its tuning curve (Sup: $R = 0.68$, p=$4.39 \times 10^{-17}$; Inf: $R = 0.68$, p=$1.24 \times 10^{-23}$) and its direction selectivity index (Sup: $R = -0.26$, p=0.005; Inf: $R = -0.38$, p=$7.10 \times 10^{-7}$). Under voltage-clamp conditions, the increment evoked greater EPSCs in Superior than Inferior oDSGCs. However, there was no difference in IPSC magnitude between cell types. Further, we found a strong correlation between the maximum firing rate of a cell to the increment and the magnitude of its peak EPSC, but not peak IPSC. These results demonstrate that Superior oDSGCs spike more than Inferior oDSGCs across multiple stimuli, and, further, that this difference in spiking is consistently associated with the amount of excitatory, but not inhibitory, input.

## Postsynaptic differences may account for shifts in E/I

Differences in the postsynaptic currents of Superior and Inferior oDSGCs could result from asymmetries in presynaptic wiring and/or the postsynaptic properties of each oDSGC type. Serial block-face electron microscopy (*Briggman et al., 2011*; *Mani et al., 2021*; *Matsumoto et al., 2019*) and analysis of dendritic stratification (*Yonehara et al., 2009*) have not yet provided evidence of presynaptic wiring differences between oDSGCs with different preferred directions. Thus, we investigated possible postsynaptic asymmetries.

We analyzed the morphology of Superior and Inferior oDSGCs by filling cells of both types with intracellular dye (*Figure 4A*). Convex hull analysis revealed that the dendritic fields of Superior oDSGCs covered a larger area than those of Inferior oDSGCs (*Figure 4B*). Sholl analysis, however, showed similar dendritic complexities (*Figure 4C*). To identify synaptic differences between cell types, we stained for the excitatory postsynaptic density scaffolding protein PSD-95 (*Koulen et al., 1998*; *Figure 4D–F*) and the inhibitory postsynaptic scaffolding protein gephyrin (*Sassoè-Pognetto et al., 1995*; *Sassoè-Pognetto and Wässle, 1997*; *Figure 4G–I*). These assays revealed no difference in the number of synaptic puncta between cell types. However, Superior oDSGCs had significantly larger excitatory, but not inhibitory, puncta (*Figure 4F, I*). This anatomy is consistent with greater amounts of excitatory synaptic input to Superior oDSGCs.

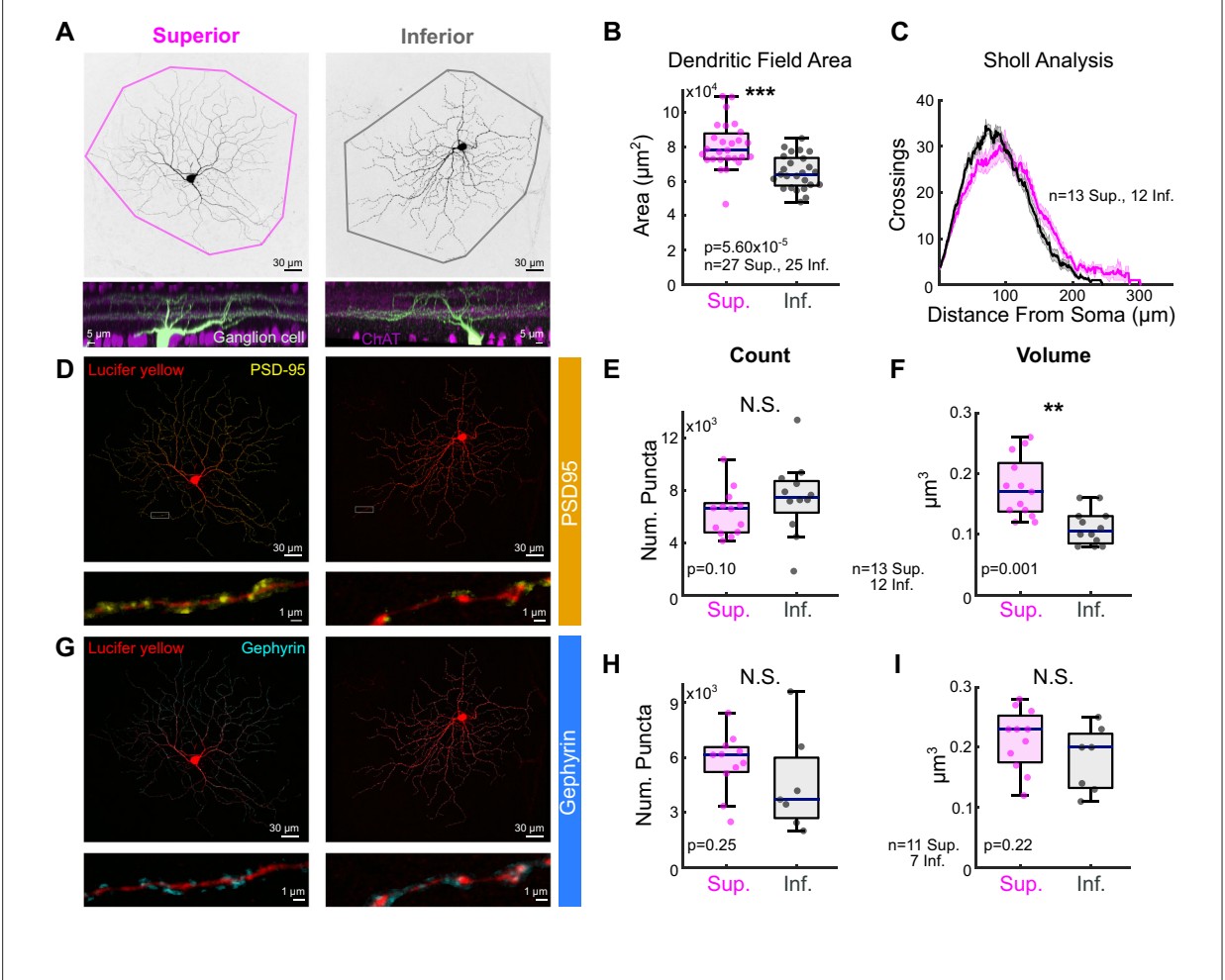

**Figure 4.** Superior ON direction-selective retinal ganglion cells (oDSGCs) have larger dendritic fields and excitatory postsynaptic sites. (**A**) Confocal images of exemplar Superior (left) and Inferior (right) oDSGCs filled with dye. Convex polygons are drawn around the tips of their dendrites. (Bottom) Side views of different Superior and Inferior oDSGCs filled and stained for acetylcholinesterase (ChAT) bands. Both cell types have dendrites that stratify in the ON and OFF ChAT bands, with the majority of dendrites in the ON sublamina. (**B**) Convex polygon areas across all filled cells. (**C**) Sholl analysis indicating the number of dendritic crossings as a function of radial distance from the soma (mean ± SEM). (**D, G**) Ganglion cells with immunostaining for (**D**) excitatory postsynaptic scaffolding protein PSD-95 or (**G**) inhibitory postsynaptic scaffolding protein gephyrin. (Bottom) Magnification of a stretch of dendrites with labeled puncta. (**E, H**) Total number of puncta within each ganglion cell for (**E**) PSD-95 and (**H**) gephyrin. (**F, I**) Quantification of average puncta volume for (**F**) PSD-95 and (**I**) gephyrin. For box plots, horizontal line represents median, box boundaries are IQR, and whiskers represent the most extreme observation within 1.5× IQR. *p<0.05, **p<0.01, ***p<0.001.

The online version of this article includes the following figure supplement(s) for figure 4:

**Figure supplement 1.** Intrinsic electrophysiological properties of ON direction-selective retinal ganglion cells (oDSGCs).

To complement these morphological observations, electrophysiological recordings revealed a number of intrinsic differences between Superior and Inferior oDSGCs. First, the membrane capacitances of Superior oDSGCs were greater than those of Inferior oDSGCs (*Figure 4—figure supplement 1A*). Sequentially recording both the spike output and EPSCs of individual oDSGCs to the full-field increment stimulus also revealed that Superior oDSGCs had lower spike-to-EPSC ratios (*Figure 3—figure supplement 2P*). This observation persisted for each direction of the drifting bar stimulus (*Figure 3—figure supplement 1D–AA*). In agreement, the input resistances of Superior oDSGCs were lower than those of Inferior oDSGCs (*Figure 4—figure supplement 1B*). These phenomena are consistent with the larger size of Superior oDSGCs relative to Inferior oDSGCs (*Figure 4B*). We found no significant differences in other intrinsic properties including the resting membrane potential and spike threshold (*Figure 4—figure supplement 1C and D*). Together, these data indicate that Superior

oDSGCs are less intrinsically excitable than Inferior oDSGCs, but that this asymmetry is outweighed by counteracting discrepancies in the magnitude of excitatory synaptic input to each cell type.

## Untuned excitation broadens spike tuning curves

That Superior oDSGCs receive relatively more excitation than Inferior oDSGCs explains their greater spike output (*Figure 2F–H*, *Figure 2—figure supplement 3A*, *Figure 3—figure supplement 2A-H*). Less obvious, however, is whether this difference in excitatory input can also account for the observation that Superior oDSGCs have wider tuning curves (*Figure 2I–K*, *Figure 2—figure supplements 2 and 3B-C*). A debate remains over whether excitatory inputs to DSGCs are directionally tuned (*Vaney et al., 2012*; *Matsumoto et al., 2019*; *Matsumoto et al., 2021*; *Percival et al., 2019*; *Poleg-Polsky and Diamond, 2011*; *Summers and Feller, 2022*; *Yonehara et al., 2013*; reviewed by *Wei, 2018*). While our results indicate that MTN-projecting RGCs might receive different amounts of excitation based on stimulus direction (*Figure 3H*), the majority of directionally tuned inputs were inhibitory (*Figure 3K*). Further, the apparent direction selectivity of EPSCs is likely partially attributable to imprecise space clamp (*Vaney et al., 2012*; *Poleg-Polsky and Diamond, 2011*; but see *Percival et al., 2019*). Thus, the extent to which the tuning curves of Superior and Inferior oDSGCs are shaped by direction-selective excitation is unclear, and we remain agnostic on this point. Instead, we focus on the more pronounced observation that Superior oDSGCs receive more excitatory input than Inferior oDSGCs across stimulus directions (*Figure 3G and I*, *Figure 3—figure supplement 1*), regardless of the extent to which this excitation is tuned. In the following experiments, we investigate the relationship between spike tuning curve shape and the overall amount of excitation to an oDSGC. We ask whether and how different magnitudes of excitatory input to Superior and Inferior oDSGCs can explain their difference in tuning curve width.

To test how the magnitude of excitation, even when directionally untuned, to an oDSGC changes the shape of its tuning curve, we measured the spikes of Superior and Inferior oDSGCs in the current-clamp configuration (*Figure 5A–B and F*) while injecting constant amounts of either positive (to add ~6 mV, 'depolarizing') or negative (to subtract ~6 mV, 'hyperpolarizing') current across stimulus directions. Importantly, the depolarizing current injections were small enough such that every cell retained a baseline firing rate of 0 Hz. This approach allowed us to investigate how providing a cell with more or less directionally untuned excitation influences the shape of its spike tuning curve, as quantified by the direction selectivity index (as in *Figure 2K*, a metric that decreases with greater tuning curve width) and the area of the normalized tuning curve (as in *Figure 2J*, referred to as 'normalized area,' a metric that increases with tuning curve width). We found that tuning curves measured under depolarizing conditions were not only larger (*Figure 5C–D and G*), but also wider, with lower direction selectivity indices (*Figure 5H*) and larger normalized areas (*Figure 5—figure supplement 1F*) than those measured under hyperpolarizing conditions. These findings are not attributable to the effects of current injection on intrinsic properties of oDSGCs (*Figure 5—figure supplement 2*). Thus, these experiments demonstrate that increasing the amount of untuned excitation to an oDSGC broadens its tuning curve.

Comparing these results to those recorded extracellularly in cell-attached recordings revealed an additional nuance: depolarizing current injections widened only the tuning curves of Inferior oDSGCs, while hyperpolarizing injections sharpened the tuning curves of both cell types (*Figure 5C and D*, *Figure 5—figure supplement 1A and B*). One possibility is that while excitation generally broadens tuning curve width, greater excitatory input minimally affects Superior oDSGCs because these cells are already positioned closer to an upper limit on this phenomenon. Nonetheless, our results highlight a causal relationship between tuning curve width and the amount of untuned excitation that an oDSGC receives.

## Spike threshold plays a dominant role in setting tuning curve width

Two complementary mechanisms could explain how untuned excitation widens tuning curves (*Figure 5E*). In the first mechanism, excitation influences the nonlinear transformation between synaptic input and spike output that is introduced by a neuron's spike threshold. More specifically, thresholding may sharpen a neuron's spike tuning curve relative to the tuning of underlying membrane fluctuations by clamping spike output at zero in response to subthreshold membrane responses that are likely to occur for null direction stimuli (*Oesch et al., 2005*). When the amount of untuned excitatory input

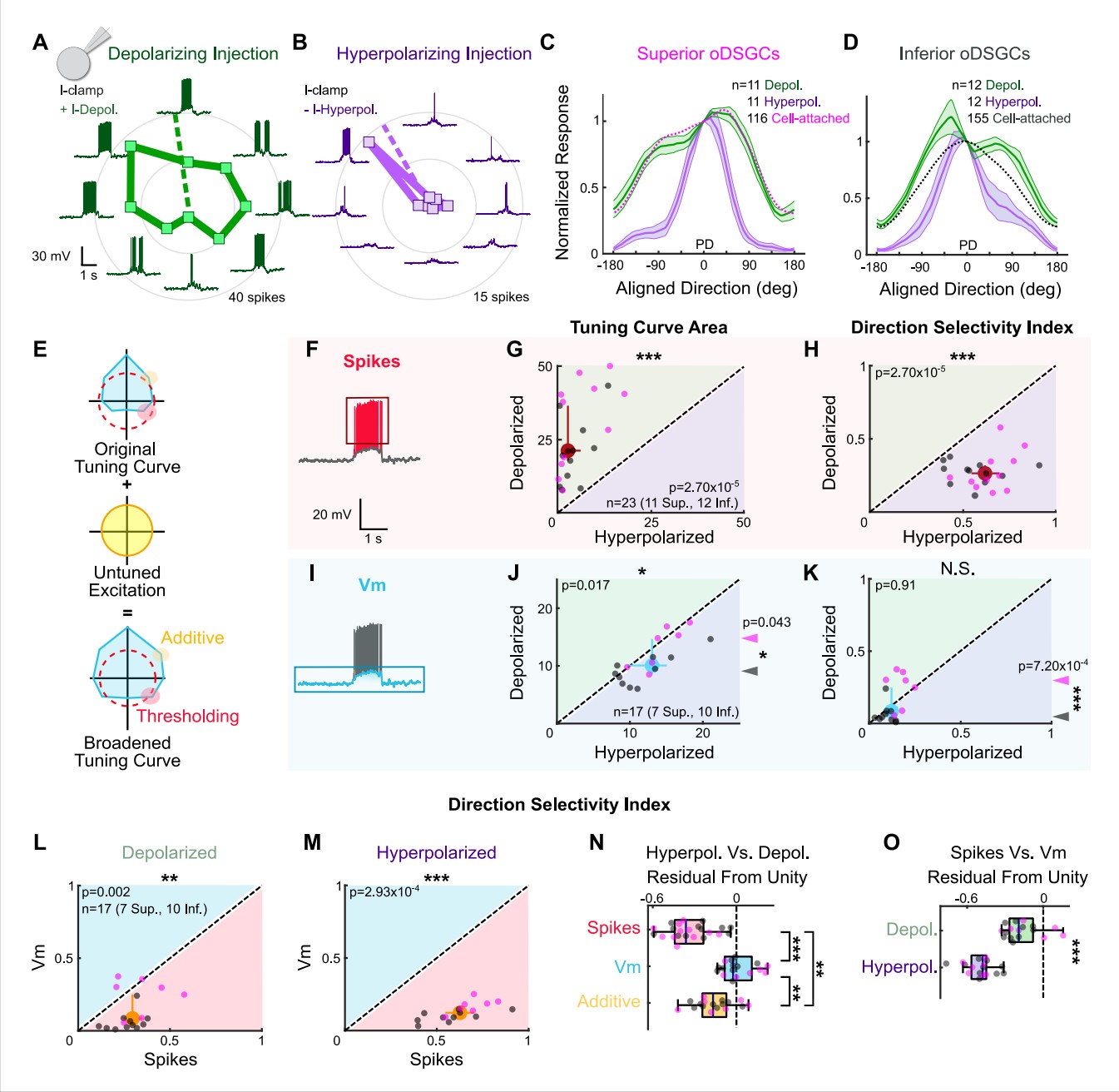

**Figure 5.** Thresholding differentiates the tuning properties of Superior and Inferior ON direction-selective retinal ganglion cells (oDSGCs). (**A, B**) Exemplar Inferior oDSGC in whole-cell current-clamp during (**A**) depolarizing and (**B**) hyperpolarizing current injection in response to a bar moving in eight directions. Numbers on concentric circles indicate spike counts. Dashed lines represent preferred directions. Coordinates are in retinal space. (**C, D**) Mean (± SEM) normalized tuning curves (aligned and normalized to the response of each cell in its preferred direction) for (**C**) Superior and (**D**) Inferior oDSGCs under conditions of depolarizing (green) and hyperpolarizing (purple) current injection. Dotted lines indicate the average normalized spike tuning curves of each population from cell-attached recordings (as in **Figure 2I**). (**E**) Illustration of the influence of untuned excitation on the tuning curve through additive (yellow) and thresholding (red) effects. The blue area indicates the membrane potential, and the dashed red line indicates the spike threshold. (**F, I**) Example whole-cell current-clamp recording in which (**F**) spikes and (**I**) subthreshold voltages (Vm) have been separated. (**G**) Linear tuning curve area and (**H**) direction selectivity index of the spike tuning curve during hyperpolarizing (abscissa) and depolarizing (ordinate) current injections. (**J**) Linear tuning curve area and (**K**) direction selectivity index of peak subthreshold membrane potential tuning curves. For (**G–H, J–K**), regions of green (or purple) indicate that the metric is greater during depolarizing (or hyperpolarizing) injections. Points that fall on the line indicate equivalent metrics under the two conditions. Individual cells are shown as dots (Superior in magenta, Inferior in gray). Large red and blue dots represent univariate medians (collapsed across cell type) and whiskers indicate 95% confidence intervals determined via bootstrapping. Significance values indicate whether the data tend to fall unevenly on one side of the unity line (two-sided signed-rank). Arrowheads in (**J, K**) represent the median

*Figure 5 continued on next page*

*Figure 5 continued*

of Superior (magenta) and Inferior (gray) oDSGCs along the unity line, and associated significance values indicate comparison between Superior and Inferior oDSGCs (two-sided rank-sum). (**L, M**) Direction selectivity index for spikes (abscissa) and simultaneously measured subthreshold voltages (ordinate) under (**L**) depolarizing and (**M**) hyperpolarizing conditions. Significance values indicate whether the data tend to fall unevenly on one side of the unity line (two-sided signed-rank). (**N**) Residuals from the unity line for individual cells from the plots in (**H**) and (**K**). Dashed line indicates unity (i.e., no difference across depolarizing and hyperpolarizing conditions). Pairwise comparisons are shown between spikes, Vm, and Vm with additive offset. (**O**) Residuals from the unity line for individual cells from the plots in (**L**) and (**M**). Dashed line indicates unity (i.e., no difference between spikes and subthreshold voltages). Comparisons are made between the depolarizing and hyperpolarizing conditions (two-sided rank-sum). For box plots, the blue line represents median, box boundaries are IQR, and whiskers represent the most extreme observation within 1.5× IQR. *p<0.05, **p<0.01, ***p<0.001.

The online version of this article includes the following figure supplement(s) for figure 5:

**Figure supplement 1.** Spike and subthreshold voltage tuning curves with directionally untuned current injections.

**Figure supplement 2.** Effects of current injection on intrinsic properties of ON direction-selective retinal ganglion cells (oDSGCs).

increases, however, membrane fluctuations more readily surpass the spike threshold for all stimulus directions, thereby broadening the spike tuning curve. In the second mechanism, untuned excitation directly circularizes the tuning curve of underlying membrane potentials by increasing null direction responses proportionally more than preferred direction responses. This 'additive' contribution would then be inherited by the spike tuning curve (*Poleg-Polsky and Diamond, 2016a*).

We first tested whether thresholding contributes to the width of oDSGC tuning curves. To do so, we isolated the underlying subthreshold voltages (Vm) from our current-clamp recordings that also contained spikes (*Figure 5I*). Three independent observations about these Vm tuning curves each suggests that thresholding critically influences spike tuning curve shape. First, unlike for spikes, depolarizing and hyperpolarizing current injections did not affect the direction selectivity (*Figure 5K and N*) or normalized area (*Figure 5—figure supplement 1G and I*) of Vm tuning curves. The total areas of Vm tuning curves were, however, slightly larger under the hyperpolarizing condition, likely due to the marginally greater driving force on excitatory conductances in this setting (*Figure 5J*). Second, the Vm tuning curves of Superior oDSGCs were larger in magnitude than those of Inferior oDSGCs (*Figure 5J* arrowheads), but also more sharply tuned, with greater direction selectivity indices (*Figure 5K* arrowheads) and smaller normalized areas (*Figure 5—figure supplement 1G* arrowheads). This latter result was anticipated from our voltage-clamp recordings that indicated that inhibition is more direction-selective in Superior oDSGCs (*Figure 3D*). It also suggests, however, that the shape of the spike tuning curve is not directly inherited from that of the underlying membrane potential and instead reflects the interplay between Vm magnitude and spike threshold. Third, spike tuning curves were more direction-selective (*Figure 5L and M*) and had smaller normalized areas (*Figure 5—figure supplement 1J and K*) than those of simultaneously measured Vms for both depolarizing and hyperpolarizing injections. The difference between the shape of the spike and Vm tuning curves was smaller for the depolarizing condition, however, because in this setting the majority of stimulus directions elicited Vm responses that surpassed the spike threshold (*Figure 5O*, *Figure 5—figure supplement 1L*). Together, these three results corroborate the notion that thresholding prominently influences the width of the spike tuning curve relative to the amount of untuned excitation that an oDSGC receives.

To test the extent to which excitation broadens oDSGC tuning curves through additive effects, we recomputed Vm tuning curves after including a constant, additive offset that reflected the average current injection supplied during depolarizing and hyperpolarizing injections. These offset-corrected Vm tuning curves were significantly wider under depolarizing conditions than they were under hyperpolarizing conditions (*Figure 5—figure supplement 1E and H*). However, the observation that (uncorrected) Vm tuning curves — in which putative additive differences *between* cell types are expected to persist — were more sharply tuned for Superior than Inferior oDSGCs (*Figure 5K*, *Figure 5—figure supplement 1G* arrowheads) indicates that thresholding has a greater influence on spike tuning curves than does additive excitation. In agreement, current injections caused significantly larger changes in spike tuning curves than in offset-corrected Vm tuning curves (*Figure 5N*, *Figure 5—figure supplement 1I*). These results demonstrate that while untuned excitation likely has some additive effect, complementary thresholding has greater influence over oDSGC spike tuning curves.

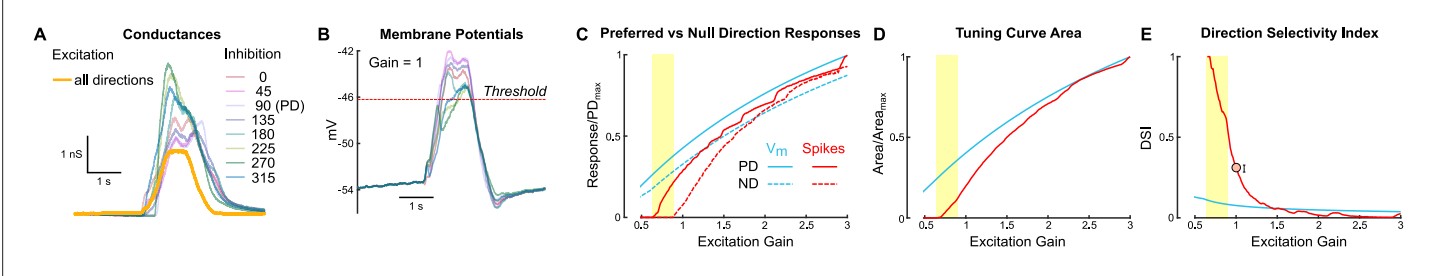

**Figure 6.** A parallel conductance model demonstrates how untuned excitation contributes to direction tuning. An exemplar ON direction-selective retinal ganglion cell (oDSGC) was modeled using parameters recorded directly from oDSGCs, including directionally tuned inhibitory conductances for each of eight drifting bar directions, and a single, untuned excitatory conductance (see 'Materials and methods'). (**A**) Inhibitory (pastel colors) and excitatory (yellow) conductances of the model oDSGC in response to bars moving in eight directions. (**B**) The parallel conductance model uses the empirically measured parameters to model the membrane potential across bar directions, shown here for the case in which the excitatory gain (i.e., a multiplication factor applied to the excitatory conductance) is set to 1.0. The red dotted line indicates the spike threshold. (**C**) Preferred (solid lines) and null (dotted lines) direction responses of peak subthreshold voltages (blue) and spikes (red) across a range of excitatory gains. Values are normalized to the maximum preferred direction response for each metric. The yellow column indicates the regime in which a null direction stimulus evokes zero spikes but a preferred direction stimulus evokes increasingly more spikes, an example of nonlinear behavior caused by the spike threshold. (**D, E**) Directional tuning properties as a function of excitatory gain for subthreshold voltages (blue) and spikes (red): (**D**) area of the linear tuning curve (normalized to the maximum area for each metric), and (**E**) direction selectivity index. Orange dot (and error bars) in (**E**) correspond to the empirically measured median (and 95% confidence interval determined via bootstrapping) direction selectivity index from cell-attached recordings, collapsed across cell types.

The online version of this article includes the following figure supplement(s) for figure 6:

**Figure supplement 1.** The normalized area of model spike tuning curves, but not subthreshold membrane potential tuning curves, is steeply influenced by excitation gain.

## A parallel conductance model recapitulates the influence of thresholding on oDSGC direction tuning

Together, our current injection experiments suggest that (1) spike thresholding plays a prominent role in setting the width of oDSGC tuning curves, and (2) additive effects contribute minorly. To independently test these findings, we built a parallel conductance model of an oDSGC based on empirically measured parameters from a separate set of cells to those used for the current injection experiments. Among these parameters were eight inhibitory conductances (one for each stimulus direction), and a single, directionally untuned excitatory conductance (*Figure 6A and B*). We then tested how manipulating the gain of the excitatory conductance affected tuning curves generated from either spikes or the peak subthreshold membrane potentials (*Figure 6C–E*, *Figure 6—figure supplement 1*; see 'Materials and methods').

Increasing the gain of untuned excitation to the model oDSGC increased the total area of both spike and Vm tuning curves (*Figure 6D*). However, while the spike tuning curve rapidly widened and became less direction-selective with increasing excitatory gain, the width of the Vm tuning curve was much less dependent on excitatory gain (*Figure 6E*, *Figure 6—figure supplement 1*). The stark difference between the spike and Vm trajectories can only be attributed to thresholding effects. On the other hand, the shallow slope of the Vm curve in *Figure 6E* reflects the additive contribution of excitation. Consistent with our physiological results, thresholding played a critical role in setting the model oDSGC's tuning curve width, whereas additive effects were relatively minor. Further, the prior observation that depolarizing current injections influenced Superior oDSGCs less than Inferior oDSGCs (*Figure 5C and D*, *Figure 5—figure supplement 1A and B*) is supported by the diminishing marginal effect of additional excitatory gain on spike tuning curve width. We also noticed that the average empirically measured direction selectivity index (*Figure 6E* circle) and normalized tuning curve area (*Figure 6—figure supplement 1*) fell within the regime where these metrics steeply depended on excitatory gain. In this regime, thresholding effects render small changes in synaptic inputs particularly influential for oDSGC direction tuning.

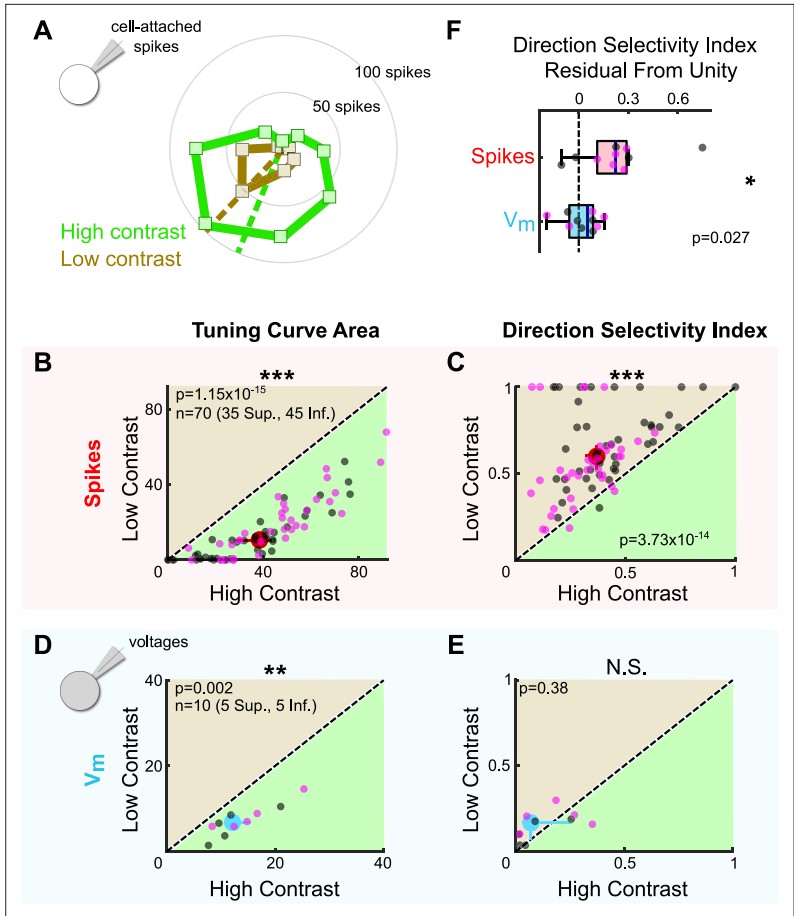

**Figure 7.** Stimulus contrast modulates the spike tuning curves of ON direction-selective retinal ganglion cells (oDSGCs). (**A**) Cell-attached tuning curves from an exemplar Superior oDSGC at high (green) and low (tan, 20% relative) contrasts. Numbers on concentric circles indicate spike counts. Dashed lines represent preferred directions. Coordinates are in retinal space. (**B**) Linear tuning curve area and (**C**) direction selectivity index from spike responses to high-contrast (abscissa) and low-contrast (ordinate) bars drifting in eight directions. Differences between Superior (magenta) and Inferior (gray) oDSGCs persist under low contrast (see *Figure 2*). (**D**) Linear tuning curve area and (**E**) direction selectivity index from peak subthreshold voltage responses to high-contrast (abscissa) and low-contrast (ordinate) bars drifting in eight directions. (**F**) Residuals from the unity line of the direction selectivity index under high- and low-contrast conditions for simultaneously measured spikes and subthreshold voltages. Comparison is made between spikes and subthreshold voltages. For all scatter plots, the region of green (or tan) indicates the metric is greater under high-contrast (or low-contrast) conditions. Points on the line indicate equivalent metrics under the two conditions. Individual cells are represented by small dots. Large dots represent univariate medians (collapsed across cell type). Whiskers indicate 95% confidence intervals determined via bootstrapping. Significance values indicate whether the data tend to fall unevenly on one side of the unity line (two-sided signed-rank). *p<0.05, **p<0.01, ***p<0.001.

The online version of this article includes the following figure supplement(s) for figure 7:

**Figure supplement 1.** Stimulus contrast modulates spike tuning curve width but not the ratio of excitation to inhibition.

**Figure supplement 2.** Two-photon targeting confirms that ON direction-selective retinal ganglion cells (oDSGCs) are contrast sensitive.

## Thresholding effects produce contrast-sensitive direction tuning in oDSGCs

The dependence of oDSGC tuning on thresholding predicts that any stimulus that influences the magnitude of synaptic inputs may also alter tuning curve shape. We tested this hypothesis by comparing the tuning curves of Superior and Inferior oDSGCs in response to high-contrast (stimuli

used in previous figures unless specified otherwise) and fivefold lower contrast (i.e., 20% relative contrast) drifting bars. For most cells, the low-contrast bars elicited fewer spikes (*Figure 7A and B*, *Figure 7—figure supplement 2A*). However, as in the case of high-contrast stimuli, Superior oDSGCs tended to spike more and have wider tuning curves than Inferior oDSGCs in response to low-contrast bars (*Figure 2H, J, and K,* insets). Nonetheless, cells of both types had sharper spike tuning curves in response to low-contrast, compared to high-contrast, stimuli (*Figure 7C*, *Figure 7—figure supplements 1A and 2B and C*). To test whether this contrast sensitivity could be attributed to thresholding, we measured the tuning curves of subthreshold membrane potentials. While the area of Vm tuning curves was greater under high-contrast conditions (*Figure 7D*), the direction selectivity (*Figure 7E*) and normalized area (*Figure 7—figure supplement 1B*) of Vm tuning curves did not change with stimulus contrast. In agreement, the fraction of cells with spike tuning curves that sharpened under low contrast was significantly different than the equivalent fraction of Vm tuning curves that followed the same trend (DSI: p=0.0046; normalized area: p=0.0019; two-sided Fisher's exact) (*Figure 7C and E*, *Figure 7—figure supplement 1A and B*), and Vm tuning curves were less affected by stimulus contrast than the tuning curves of simultaneously measured spikes (*Figure 7F*, *Figure 7—figure supplement 1C*). These results indicate that thresholding is critical to setting oDSGC spike tuning curve width across stimulus contrasts, just as it is across cell types.

The finding that oDSGCs are contrast-sensitive departs from previous work which has shown that direction tuning in ooDSGCs is contrast-invariant (*Poleg-Polsky and Diamond, 2016b*; *Sethuramanujam et al., 2016*; *Sethuramanujam et al., 2017*). Contrast invariance in ooDSGCs appears to rely on the stability of E/I across contrasts. For this reason, we tested whether the contrast sensitivity of oDSGCs was associated with a mutable E/I by sequentially measuring excitatory and inhibitory synaptic currents for preferred and null direction stimuli across contrasts. Our data show that E/I in neither the preferred nor null direction changed systematically with contrast (*Figure 7—figure supplement 1D and E*). Further, the fraction of cells with spike tuning curves that sharpened under low-contrast conditions was significantly greater than the fraction of E/I ratios that were lower under low-contrast conditions (DSI: PD p=0.0044, ND p=0.0044; normalized area: PD p=0.0132, ND p=0.0132; two-sided Fisher's exact) (*Figure 7C*, *Figure 7—figure supplement 1A, D, and E*). These results suggest that as in ooDSGCs, E/I in oDSGCs is relatively stable across contrasts. However, the spike tuning curves of oDSGCs are nonetheless contrast-sensitive. Thus, stable E/I alone is insufficient to maintain the contrast invariance of spikes. In the case of oDSGCs, changes to the absolute magnitude of synaptic inputs across contrasts, along with thresholding effects, appear to trump the contrast invariance of E/I.

To further test the extent to which the magnitude of synaptic inputs can affect oDSGC tuning curves even with stable E/I, we revisited our parallel conductance model of an exemplary oDSGC. We modeled oDSGC responses while changing the gain of excitatory and inhibitory conductances together, thereby keeping E/I constant while nevertheless adjusting the total magnitude of E and I. As in our empirical data, lowering the gain of E and I together to simulate responses to low-contrast stimuli resulted in sharper spike tuning curves. The width of Vm tuning curves, however, remained relatively contrast invariant (*Figure 7—figure supplement 1F*). These results recapitulate our empirical findings and indicate that the spike threshold nonlinearity influences spike tuning curve width across contrasts. Other stimulus parameters that affect the magnitude of excitation, inhibition, or both are also likely to modulate the direction tuning properties of oDSGCs.

## A subtraction algorithm predicts vertical OKR from oDSGC activity

Having established the tuning properties of Superior and Inferior oDSGCs, we asked whether the asymmetries between these cell types, along with their contrast-sensitivities, could explain how vertical OKR changes across stimulus conditions. Cross-species work has established that Superior and Inferior oDSGCs are likely centrally integrated by a subtraction algorithm (*Simpson, 1984*; *van der Togt et al., 1993*; *Soodak and Simpson, 1988*). In this model, OKR is predicted on the basis of the *difference* in spike rate between Superior and Inferior oDSGCs rather than by the absolute spike rate of either cell type. A number of observations support this model: (1) stimuli that activate both Superior and Inferior oDSGCs (e.g., a full-field increment of light) do not elicit OKR, (2) stimuli that differentially activate Superior and Inferior oDSGCs (e.g., drifting gratings) maximally drive OKR (*Figures 1 and 2*), (3) Superior and Inferior oDSGCs project to separate MTN subnuclei (*Yonehara*

*et al., 2009*; *van der Togt et al., 1993*) that are mutually connected by inhibitory interneurons (i.e., a simple differentiating circuit) (*van der Togt et al., 1993*; *Giolli et al., 1985*), (4) MTN neurons prefer either superior or inferior motion (*Yonehara et al., 2009*; *van der Togt et al., 1993*; *Soodak and Simpson, 1988*; *Grasse and Cynader, 1982*; *Natal and Britto, 1988*; *Simpson et al., 1979*), but their preferred and null directions are not 180° apart; instead, they correspond to the preferred directions of opposing oDSGC types (*Soodak and Simpson, 1988*; *Natal and Britto, 1988*; *Simpson et al., 1979*), which may differ by less than 180° (*Oyster and Barlow, 1967*; but see *Sabbah et al., 2017*), and (5) MTN neurons maintain moderate baseline spike rates that are both augmented by preferred direction stimuli and diminished by null direction stimuli (*Soodak and Simpson, 1988*; *Grasse and Cynader, 1982*; *Natal and Britto, 1988*; *Simpson et al., 1979*). Together, along with the simplicity of the vertical OKR pathway and its isolation from other visual circuits, these lines of evidence all point to a circuit motif in which MTN neurons encode the difference in spike rate between Superior and Inferior oDSGCs. Thus, superior OKR likely occurs when Superior oDSGCs spike sufficiently more than Inferior oDSGCs, and vice versa. The robustness of OKR is putatively related to the magnitude of the spike rate difference (*Figure 8A*).

We used our empirically recorded electrophysiology data from Superior and Inferior oDSGCs to generate hypotheses for how OKR gain would change across stimulus conditions. Superior and Inferior oDSGC firing rate distributions were compared for superior and inferior drifting bars at high (full) and low (20% relative) contrast (i.e., four stimulus conditions) (*Figure 8B*, *Figure 8—figure supplement 1B–E*). We inferred that the relative OKR gain under each stimulus condition would be related to the corresponding difference in spike rate between Superior and Inferior oDSGCs under that same condition. For instance, gain in response to a high-contrast superior grating was predicted by the difference between the preferred direction responses of Superior oDSGCs and the null direction responses of Inferior oDSGCs to high-contrast bars. Importantly, such inferences constitute linear predictions of gain. Allowing for the possibility that downstream circuitry incorporates additional monotonic nonlinearities, a linear prediction is consistent with behavior so long as it predicts gain changes in the correct *direction* (but not magnitude) across stimulus conditions.

The asymmetries in the tuning curves of Superior and Inferior oDSGCs resulted in the following key predictions for how OKR gain would change across stimulus directions and contrasts: (1) gain would decrease with stimulus contrast, (2) OKR would be asymmetric, with superior stimuli eliciting greater gain than inferior stimuli, and (3) this asymmetry between responses to superior and inferior stimuli would decrease with stimulus contrast (*Figure 8C*, *Figure 8—figure supplement 1*). Next, we tested these predictions in behaving mice. OKR in the superior and inferior directions was measured in response to high-contrast (full) and low-contrast (20% relative) oscillating gratings. All of the linear predictions were consistent with behavior (*Figure 8H*, *Figure 8—figure supplements 1 and 3*). Most notably, gain decreased with stimulus contrast (*Figure 8H*, *Figure 8—figure supplement 1H, I, and P*, *Figure 8—figure supplement 3*), and the asymmetry between superior and inferior OKR that we originally noticed under high-contrast conditions diminished in response to low-contrast stimuli (*Figure 8H*, *Figure 8—figure supplement 1L, M, and Q*, *Figure 8—figure supplement 3*). While these results may be related, they do not *necessitate* each other, and instead rely on further subtleties in the relationship between Superior and Inferior oDSGC responses at both contrasts. Indeed, permuting the behavioral predictions by scrambling which cellular responses were assigned to superior/inferior motion and high/low contrast revealed that only five permutations (out of 256 possibilities, 1.95%) accurately matched our behavioral results. Consistent with these findings, the relationship between our linear predictions and behavioral results was fit well by a monotonic function that was near linear within the measured regime (*Figure 8D*).

Finally, we tested whether instantaneous subtraction of Superior and Inferior oDSGC firing rates on millisecond timescales could also predict vertical OKR behavior. We directly recorded the spikes of Superior and Inferior oDSGCs in response to oscillating gratings with the same parameters as those used to induce behavioral OKR. This stimulus evoked more spikes in Superior oDSGCs as gratings drifted dorsal to ventral on the retina (superior motion), and more spikes in Inferior oDSGCs as gratings drifted ventral to dorsal on the retina (inferior motion) (*Figure 8E*, *Figure 8—figure supplement 2C and D*). To make behavioral predictions, the average population instantaneous firing rates of Superior and Inferior oDSGCs were subtracted every 5 ms over the course of an oscillation cycle (*Figure 8—figure supplement 2E and F*). Such values constituted linear predictions of instantaneous

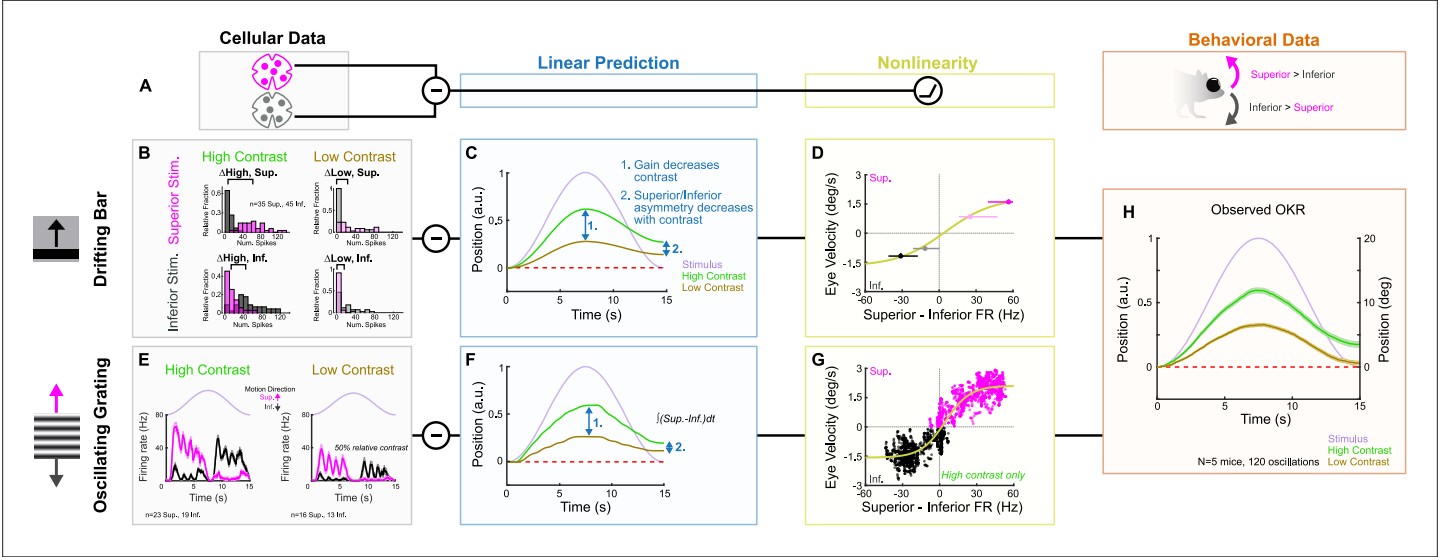

**Figure 8.** ON direction-selective retinal ganglion cell (oDSGC) responses predict the optokinetic reflex (OKR) across stimulus types, directions, and contrasts. (**A**) Schematic of the putative computation between oDSGCs and OKR, consisting of a subtraction between Superior and Inferior oDSGC spikes and a nonlinearity. (**B–H**) Two separate implementations of the subtraction model described in (**A**). (**B–D**) Prediction of OKR behavior from oDSGC spike responses to the drifting bar stimulus. (**B**) Distributions of Superior (magenta) and Inferior (gray) oDSGC spike responses across high-contrast (left) and low-contrast (right, 20% relative) superior (top) and inferior (bottom) drifting bars. The brackets denote the difference between the medians of the Superior and Inferior oDSGC response distributions in each condition. (**C**) Linear predictions of OKR are made for the average eye velocity over the course of each half oscillation cycle based on the difference in firing rate between Superior and Inferior oDSGCs (i.e. brackets in [**B**]) to high- and low-contrast bars drifting in the corresponding stimulus direction. The shape of the curves as sinusoids is inferred from the stimulus position over time (lavender). (**D**) The empirically computed nonlinearity shows the relationship between linear behavioral predictions (as in [**C**]) from the drifting bar stimulus and the corresponding average eye velocities measured during behavioral OKR experiments for superior (magenta points) and inferior (gray points) stimuli at high (dark points) and low (light points) contrast. The points indicate univariate medians for each condition and whiskers are 95% confidence intervals computed via bootstrapping (vertical error bars are too small to see). The solid line is a fitted sigmoid function of the form described in **Equation 6** and has parameters $v_{min} = -1.69$, $v_{max} = 1.82$, $r_{50} = 4.93$, $m = 0.022$. (**E–G**) Prediction of OKR behavior from oDSGC responses to an oscillating sinusoidal grating. (**E**) Median responses of Superior (magenta) and Inferior (gray) oDSGCs across a single cycle of the oscillating grating for high-contrast (left) and low-contrast (right, 50% relative) stimuli. Lavender traces represent the relative position of the stimulus across time. Directions of superior/inferior motion are in visual space and indicated by arrows. These recordings were made using two-photon targeting. (**F**) Linear predictions of OKR across time made by subtracting the instantaneous firing rates of Superior and Inferior oDSGCs to generate predictions for instantaneous eye velocity, and integrating to predict eye position. (**G**) The empirically computed nonlinearity shows the difference between Superior and Inferior oDSGC firing rates plotted against the time-matched average eye velocity of behaving animals. Each small point represents the average firing rate difference and eye velocity at a single time point over the course of one stimulus oscillation cycle. Magenta points represent superior eye velocities (i.e., above 0 on the ordinate), and gray points represent inferior eye velocities (i.e., below 0 on the ordinate). The solid line is a fitted sigmoid curve of the form described in **Equation 6** and has parameters $v_{min} = -1.57$, $v_{max} = 2.11$, $r_{50} = 4.61$, and $m = 0.044$. Only data for the high-contrast condition were used. (**H**) Mean eye position of behaving mice in response to a single oscillation of a high-contrast (green, as in **Figure 1I**) or low-contrast (tan, 20% relative) gratings. Compare to linear predictions in (**C**) and (**F**).

The online version of this article includes the following figure supplement(s) for figure 8:

**Figure supplement 1.** Behavioral prediction for the optokinetic reflex (OKR) from spike responses to the drifting bar stimulus.

**Figure supplement 2.** ON direction-selective retinal ganglion cell (oDSGC) responses to oscillating gratings.

**Figure supplement 3.** The optokinetic reflex (OKR) at low contrast.

**Figure supplement 4.** Baseline vertical eye movements to low-contrast stimuli (see also **Figure 1—figure supplement 2**).

eye velocity for each time point, and their cumulative integrals yielded predictions of eye position (**Figure 8F**, **Figure 8—figure supplement 2G and H**). Recordings were initially made in response to both high (full) and 20% relative contrast gratings; however, the low-contrast condition failed to evoke consistent spiking in oDSGCs. Thus, we instead used high (full) and 50% relative contrast gratings to predict how responses would change across contrasts (see 'Materials and methods'). Moment-by-moment subtraction of oDSGC firing rates indicated that (1) eye movements would track the sinusoidal pattern of stimulus motion, (2) gain would decrease with decreasing stimulus contrast, (3) OKR would be asymmetric with greater gain in response to superior motion than inferior motion, and (4)

asymmetries between Superior and Inferior OKR would decrease with decreasing stimulus contrast. Not only do these predictions match those generated from the drifting bar stimulus, but they also accurately predict OKR in behaving mice (*Figure 8H*). The moment-by-moment difference between Superior and Inferior oDSGC firing rates was also generally linearly related to the time-matched eye velocity of behaving animals, with nonlinear regimes occurring only at extreme firing rate differences (*Figure 8G*). Together, these results provide a neurophysiological explanation for how vertical OKR changes across multiple stimulus conditions and reveal that the circuit and cellular properties that shape oDSGC motion encoding have direct and predictable consequences for behavior.

## Discussion

Our results depict a neurophysiological mechanism by which vertical OKR changes across stimulus conditions. We demonstrate that superior and inferior OKR are asymmetric in adult mice (*Figure 1*), and show how this behavioral phenomenon can be traced to novel asymmetries in the direction tuning properties of Superior and Inferior oDSGCs (*Figure 2*). Mechanistically, a shift in the balance of excitatory and inhibitory inputs across cell types influences direction tuning, primarily through an effect associated with spike thresholding (*Figures 3–6*). Similar thresholding effects also confer contrast-sensitivity of the spike tuning curve, even when E/I is contrast-invariant (*Figure 7*). Together, these cellular properties accurately predict how vertical OKR changes with stimulus direction and contrast (*Figure 8*).

Directional asymmetries in OKR are common across species. Besides the vertical asymmetries investigated here, horizontal OKR is asymmetric in many organisms (*Masseck and Hoffmann, 2009*; *Mowrer, 1936*). This asymmetry manifests nearly universally as higher OKR gain in response to temporal-to-nasal (anterior) motion than to nasal-to-temporal (posterior) motion, and often only when stimuli are viewed monocularly. Such horizontal asymmetries may similarly be linked to direction tuning in the retina: while anterior-preferring oDSGCs are critical to horizontal OKR (*Dhande et al., 2015*; *Osterhout et al., 2015*; *Yonehara et al., 2016*), posterior-preferring oDSGCs were only recently identified in rodents and display distinct direction tuning properties compared to their anterior-preferring counterparts (*Sabbah et al., 2017*). Further, a subtraction mechanism between horizontally tuned oDSGCs may also underlie horizontal OKR (*Hoffmann and Fischer, 2001*). However, at least two confounds obscure a connection between the tuning properties of anterior and posterior oDSGCs and asymmetries in horizontal OKR. First, oDSGCs in the left and right eyes, and their contralateral central targets (NOT/DTN), encode different absolute directions of stimulus motion (reflection occurs over the sagittal body axis). To compensate, signals are compared across eyes/hemispheres prior to behavior (*Hoffmann and Fischer, 2001*), and, in some species, NOT/DTN receives descending, often binocular, inputs from visual cortex (*Masseck and Hoffmann, 2009*; *Giolli et al., 2006*; *Liu et al., 2016*; *Wood et al., 1973*). Second, recent studies have suggested that ooDSGCs could be involved in horizontal OKR (*Dhande et al., 2013*; *Kay et al., 2011*). Asymmetries in the horizontal version of the behavior may thus depend on more complex considerations.

Analogous confounds are less concerning when considering the mechanisms that could account for asymmetries in vertical OKR. For one, stimuli that induce vertical OKR, such as the gratings used here, are perceived identically by both eyes. Interhemispheric communication is unlikely to influence behavior under such conditions. Though signals are probably also exchanged between the horizontal and vertical OKR pathways (*Simpson, 1984*; *Giolli et al., 2006*; *Lilley et al., 2019*), these channels may play a minimal role in shaping OKR to purely vertical stimuli. Finally, while it has been suggested that a vertically tuned ooDSGC may also project to MTN (*Kay et al., 2011*), this cell type was not clearly revealed by either anatomical or electrophysiological analyses during our retrograde labeling experiments (*Figure 2—figure supplement 1*, *Figure 3—figure supplement 2*). Instead, we find that two populations of ganglion cells project to MTN, and that these cells can be classified as Superior and Inferior oDSGCs. These results are consistent with characterizations of MTN-projecting RGCs across many species (*Yonehara et al., 2008*; *Yonehara et al., 2009*; *Cook and Podugolnikova, 2001*; *Dann and Buhl, 1987*; *Ruff et al., 2021*). Thus, unlike for horizontal OKR, asymmetries in vertical OKR can be explained more simply by the physiology of oDSGCs.

Despite the possibility that asymmetries in vertical and horizontal OKR are influenced by separate mechanisms, the two phenomena could share a common ethological function. Optic flow associated with forward locomotion typically includes a large posterior component. Thus, it has been suggested

that posterior OKR is less reliable than anterior OKR in order to mitigate lateral eye movements that might otherwise occur when an animal walks forward (*Tauber and Atkin, 1968*). Similar reasoning may explain the asymmetry between superior and inferior OKR (*Takahashi et al., 1978*; *Grasse and Cynader, 1988*; *Takahashi and Igarashi, 1977*). Indeed, recent work in freely moving mice demonstrated that optic flow has a stronger inferior than superior component (*Holmgren et al., 2021*). Thus, the cellular and behavioral asymmetries identified in this study may provide an ethological advantage by mitigating aberrant eye movements during forward locomotion. Future work should address this possibility by mapping OKR gain as a function of the extent to which various stimuli reflect locomotion-associated optic flow.

From a physiological perspective, our results provide the first evidence that Superior and Inferior oDSGCs encode motion asymmetrically. Superior oDSGCs produce more spikes and have broader tuning curves than Inferior oDSGCs (*Figure 2*). These findings coincide with previous work demonstrating genetic (*Al-Khindi et al., 2022*) and anatomical differences between oDSGC types. Multiple transgenic lines are known to label either Superior or Inferior oDSGCs, but not both (*Lilley et al., 2019*; *Yonehara et al., 2008*; *Yonehara et al., 2009*; *Ruff et al., 2021*). The axons of Superior and Inferior oDSGCs take separate retinofugal tracts and project to different MTN subnuclei in mice (*Yonehara et al., 2009*). Additional differences exist between vertically and horizontally tuned oDSGCs (*Dhande et al., 2015*; *Osterhout et al., 2015*). The competitive advantage of mitigating OKR during forward locomotion may have thereby shaped the development of many differences between oDSGC types.

The asymmetric spike tuning of Superior and Inferior oDSGCs is associated with differences in the magnitude of excitatory synaptic input to each cell type (*Figure 3*). Three synaptic partners are primary candidates for the source of this asymmetry: (1) glutamatergic input from bipolar cells (types 5 and 7), (2) cholinergic input from SACs, and (3) glutamatergic input from VGluT3 amacrine cells (*Amthor et al., 2002*; *Yonehara et al., 2011*; *Yoshida et al., 2001*; *Mani et al., 2021*; *Matsumoto et al., 2019*; *Yonehara et al., 2013*; *Lee et al., 2014*; *Sivyer et al., 2019*). Glutamatergic conductances in oDSGCs rely on AMPARs and NMDARs, whereas cholinergic conductances rely on nicotinic AChRs (*Kittila and Massey, 1997*). These conductances have been studied primarily in ooDSGCs and deserve further characterization in oDSGCs. Interestingly, VGluT3 amacrine cells also corelease glycine, which cancels oDSGC spiking in response to high-velocity stimuli (*Mani et al., 2021*; *Summers and Feller, 2022*; *Lee et al., 2014*; *Sivyer et al., 2019*). Asymmetries at the VGluT3-oDSGC synapse could coincide with differences in the speed tuning properties of Superior and Inferior oDSGCs. A combination of genetic, optical, and pharmacological manipulations could distinguish among these possible sources of asymmetric tuning.

In addition to the gain of excitatory inputs, other mechanisms could contribute to tuning curve differences between Superior and Inferior oDSGCs. Debate remains over the extent to which excitatory inputs to DSGCs are directionally tuned (*Vaney et al., 2012*; *Matsumoto et al., 2019*; *Matsumoto et al., 2021*; *Percival et al., 2019*; *Poleg-Polsky and Diamond, 2011*; *Summers and Feller, 2022*; *Yonehara et al., 2013*). Our data indicate that excitation may be less direction-selective in Superior than in Inferior oDSGCs (*Figure 3H*), which could contribute to their broader spike tuning curves. Qualitatively, excitation also showed a bimodal average tuning curve in Superior oDSGCs (*Figure 3G*) that matched their average spike tuning curve (*Figure 2G*). However, bimodal spike tuning curves also resulted from directionally untuned depolarizing current injections in both cell types (*Figure 5C and D*), so the influence of excitatory tuning remains unclear. In addition, previous studies with simultaneous somatic and dendritic recordings have revealed that dendritic spikes in rabbit oDSGCs contribute to directional tuning (*Sivyer and Williams, 2013*). Different spatial distributions of voltage-gated sodium channels along dendrites could also contribute to asymmetric direction tuning between oDSGC types. Indeed, we show that Superior and Inferior oDSGCs have distinct morphologies (*Figure 4*). Thus, while a single-compartment conductance model captured the empirical data in this study (*Figure 6*), development of multicompartment models could elucidate potential contributions of dendritic spikes to asymmetric tuning between oDSGC types. Intriguingly, however, among mechanisms that are unlikely to explain differences between Superior and Inferior oDSGC tuning curves is that of direction-selective inhibition. We find that inhibition is more sharply tuned in Superior oDSGCs (*Figure 3C and D*), and that this is associated with sharper Vm tuning curves (*Figure 5K*, arrowheads). In agreement, analyses leveraging our parallel conductance model also demonstrated that, all else equal, sharper inhibitory tuning contributes to sharper Vm and spike

tuning in oDSGCs (not shown). Nonetheless, empirically measured spike tuning curves are broader in Superior oDSGCs than in Inferior oDSGCs (*Figure 2J and K*). Thus, while the relationship between inhibitory tuning curve shape and spike tuning curve shape is nuanced and requires further investigation, the difference in spike tuning curve shape between Superior and Inferior oDSGCs is unlikely to be explained by inhibitory tuning. Evidently, other mechanisms, including differences in excitatory gain (*Figure 3G*), counteract and outweigh the influence of differences in inhibition. Consequently, a comprehensive understanding of direction selectivity will require mapping the contributions of mechanisms that have been less well-studied than directionally tuned inhibition.

Further insight into how oDSGCs encode motion can be gained by comparing their tuning properties to those of the more comprehensively studied ooDSGCs. While prior work has focused on speed tuning as the primary difference between oDSGCs and ooDSGCs (*Oyster, 1968*; *Oyster et al., 1972*; *Mani et al., 2021*; *Summers and Feller, 2022*; *Sivyer et al., 2019*), our results reveal an additional difference between these two classes of DSGCs that has been previously overlooked: oDSGCs are contrast sensitive (*Figure 7*), whereas ooDSGCs are not (*Sethuramanujam et al., 2016*). In ooDSGCs, preservation of E/I across contrasts is apparently critical for retaining contrast-invariant direction tuning (*Poleg-Polsky and Diamond, 2016b*; *Sethuramanujam et al., 2017*). Counterintuitively, we find that E/I in oDSGCs is also relatively stable across contrasts (*Figure 7—figure supplement 1D and E*). Therefore, stability of E/I is not alone sufficient for contrast-invariant spike tuning; independence from thresholding effects is also required. Indeed, thresholding modulates oDSGC direction tuning following changes to either E/I (*Figures 5–6*) or the absolute magnitude of E and I (*Figure 7—figure supplement 1F*). Evidently, the influence of thresholding constitutes a major difference in the mechanisms that govern how oDSGCs and ooDSGCs encode motion.

That E/I is contrast-invariant in both oDSGCs and ooDSGCs also indicates some extent of shared circuitry between these cell types. Contrast invariance of E/I in ooDSGCs relies on postsynaptic NMDA conductances and constrains possible presynaptic wiring motifs (*Poleg-Polsky and Diamond, 2016b*; *Sethuramanujam et al., 2017*). oDSGCs likely share many of these features. Indeed, serial block-face electron microscopy has confirmed that oDSGCs and ooDSGCs share many of their presynaptic partners (*Briggman et al., 2011*; *Ding et al., 2016*; *Mani et al., 2021*; *Matsumoto et al., 2019*; *Matsumoto et al., 2021*). Nonetheless, differences in the intrinsic properties between and within DSGC classes, including dependence on thresholding, could magnify the impact of subtle circuit differences on spike output.

Finally, our results show that vertical OKR is predicted by a simple subtraction between the outputs of Superior and Inferior oDSGCs (*Figure 8*). It is interesting that asymmetries in oDSGCs are not apparently corrected by downstream circuitry, considering that normalization operations pervade the nervous system (*Carandini and Heeger, 2011*). One explanation is that there is no simple compensatory solution to normalize the responses of Superior and Inferior oDSGCs because multiple stimulus parameters (e.g., stimulus direction and contrast) simultaneously affect the asymmetry magnitude. On the other hand, the ethological advantage of asymmetric OKR (i.e., mitigating aberrant eye movements during forward locomotion) may have provided sufficient evolutionary pressure to allow asymmetries between Superior and Inferior oDSGCs to propagate to behavior when they might otherwise have been compensated. Regardless, a linear subtraction of oDSGC outputs offers an accurate and parsimonious explanation of vertical OKR. Moreover, this algorithm fits well with both the anatomy and physiology of MTN and the isolation of the vertical OKR pathway from other visual circuits (*Giolli et al., 2006*; *Yonehara et al., 2009*; *van der Togt et al., 1993*; *Soodak and Simpson, 1988*; *Giolli et al., 1985*; *Grasse and Cynader, 1982*; *Natal and Britto, 1988*; *Simpson et al., 1979*).

Similar subtraction algorithms are prevalent across the animal kingdom. In the mammalian retina, such computations confer both the spatial center-surround (*Barlow, 1953*; *Hartline et al., 1952*; *Kuffler, 1953*) and chromatic dichotomy (*Dacey et al., 2014*; *De Monasterio and Gouras, 1975*; *Field et al., 2009*) of receptive fields. In *Drosophila*, spatially offset antennae allow accurate estimation of wind velocity by differentiation of signals across two input sites (*Suver et al., 2019*). Similar mechanisms likely underlie tropotaxic orienting behaviors that also rely on spatially offset receptors, including arthropod antennae (*Martin, 1965*), reptile forked tongues (*Schwenk, 1994*), and mammalian ears (*Middlebrooks and Green, 1991*). The ubiquity of this circuit motif may reflect an efficient solution for integrating complementary information streams. Our results highlight how such circuits can be influenced by subtle asymmetries across input channels. Further investigation will determine

whether diverse sensory systems rely on asymmetric inputs to adaptively change behavior across stimulus conditions.

# Materials and methods

**Key resources table**

| Reagent type (species) or resource | Designation | Source or reference | Identifiers | Additional information |
|---|---|---|---|---|
| Strain, strain background (*Escherichia coli*) | WT C57BL/6J Mice | The Jackson Laboratory | N/A | |
| Antibody | Anti-Lucifer yellow (rabbit polyclonal) | Invitrogen | Cat #A5750; RRID:AB_1501344 | IF (1:500) |
| Antibody | Anti-gephyrin (mouse monoclonal) | Synaptic System | Cat# 147111; RRID:AB_887719 | IF (1:500) |
| Antibody | PSD-95 MAGUK scaffolding protein (mouse monoclonal) | Neuromab | Cat# 75-028; RRID:AB_2292909 | IF (1:500) |
| Antibody | Anti-choline acetyltransferase (goat polyclonal) | Millipore | Cat# AB144P; RRID:AB_2079751 | IF (1:500) |
| Antibody | Anti-mouse-Dylight 405 (donkey polyclonal) | Jackson ImmunoResearch | Cat# 715-475-150; RRID:AB_2340839 | IF (1:1000) |
| Antibody | Anti-mouse-Alexa 647 (donkey polyclonal) | Jackson ImmunoResearch | Cat# 715-605-151; RRID:AB_2340863 | IF (1:1000) |
| Antibody | Anti-mouse IgG, Fc_Subclass 1 Specific-Dylight 405 (goat polyclonal) | Jackson ImmunoResearch | Cat# 115-475-205; RRID:AB_2338799 | IF (1:1000) |
| Antibody | Anti-mouse moncolonal IgG, Fc_Subclass 2a Specific-Alexa 647 (goat polycolonal) | Jackson ImmunoResearch | Cat# 115-605-206; RRID:AB_2338917 | IF (1:1000) |
| Antibody | Anti-rabbit-Alexa 488 (donkey polyclonal) | Jackson ImmunoResearch | Cat# 711-545-152; RRID:AB_2313584 | IF (1:1000) |
| Antibody | Anti-goat IgG (H+L)-Alexa 647 (donkey polyclonal) | Jackson ImmunoResearch | Cat# 705-605-147; RRID:AB_2340437 | IF (1:1000) |
| Antibody | Streptavidin 488 conjugate antibody | Molecular Probes | Cat# S32354; RRID:AB_2315383 | IF (1:400) |
| Chemical compound, drug | Ames' Medium | United States Biological | Cat# A1372-25 | |
| Chemical compound, drug | Vectashield | Vector Laboratories | Cat# H-1000; RRID:AB_2336789 | |
| Chemical compound, drug | Red Retrobeads | Lumafluor | https://lumafluor.com/information | |
| Chemical compound, drug | Lucifer yellow CH dilithium salt | Sigma | Cat# L0259 | |
| Chemical compound, drug | Biocytin | Invitrogen | Cat# B1592 | |
| Software, algorithm | Amira | Thermo Fisher Scientific | https://www.fei.com/software/amira-avizo/; RRID:SCR_014305 | |
| Software, algorithm | Bassoon | *Harris, 2022* | https://doi.org/10.5281/zenodo.6757605; RRID:SCR_023333 | |
| Software, algorithm | Igor Pro | Igor Pro | RRID:SCR_000325 | |
| Software, algorithm | ImageJ | NIH | https://imagej.nih.gov/ij/; RRID::SCR_003070 | |
| Software, algorithm | Imaris | Bitplane | http://www.bitplane.com/; RRID:SCR_007370 | |

*Continued on next page*

Continued

| Reagent type (species) or resource | Designation | Source or reference | Identifiers | Additional information |
|---|---|---|---|---|
| Software, algorithm | MATLAB | MathWorks | https://www.mathworks.com/products/matlab.html; RRID:SCR_001622 | |
| Software, algorithm | Meshmapper | Paul Bourke | http://paulbourke.net/dome/meshmapper/ | |
| Software, algorithm | ObjectFinder | *Della Santina et al., 2013* | https://github.com/lucadellasantina/ObjectFinder; https://zenodo.org/record/4767847; RRID:SCR_023319 | |
| Software, algorithm | Psychopy | Open Science Tools Ltd. *Peirce, 2007* | https://psychopy.org/about/index.html | |
| Software, algorithm | ScanImage | MBF Bioscience | https://www.mbfbioscience.com/products/scanimage | |
| Software, algorithm | StreamPix | NorPix | https://www.norpix.com/products/streampix/streampix.php | |
| Software, algorithm | Symphony and Stage | *Cafaro, 2019* | https://github.com/Symphony-DAS/symphony-matlab; https://github.com/Stage-VSS/stage-v1 | |
| Software, algorithm | VolumeCut | *Della Santina et al., 2021; Della Santina, 2021* | https://github.com/lucadellasantina/VolumeCut; https://doi.org/10.5281/zenodo.5048331 | |

## Resource availability

### Lead contact
Further information and requests for resources and reagents should be directed to and will be fulfilled by the lead contact, Felice Dunn (Felice.Dunn@ucsf.edu).

### Material availability
This study did not generate new unique reagents.

## Experimental model and subject details

### Animals
Adult wildtype C57BL/6 mice between the ages of postnatal day P60 and P100 of both sexes were used for all experiments. Animals were kept on a 12 hr dark–12 hr light cycle with continuous access to food and water. All experiments were performed in accordance with protocols approved by the University of California, San Francisco Institutional Animal Care and Use Program.

## Method details

### Behavior rig
To accurately evoke and measure OKR, we custom-designed a behavior rig that was capable of presenting full-field, binocular stimuli to behaving mice. The design of the rig was based on *Denman et al., 2017*. Briefly, an acrylic hemisphere (diameter = 24 inches, California Quality Plastics) was covered with a custom paint that had 50% diffuse reflectivity between 350 and 750 nm (Twilight Labs) in order to limit reflections within the hemisphere. Stimuli were emitted from a DLP projector with peak emission at 405 nm (LightCrafter through EKB Technologies) and were reflected onto the hemisphere via a silver-coated brass hemisphere ('convex mirror,' diameter = 6 inches, Wagner). Stimuli were built using Psychopy (*Peirce, 2007*) (https://www.psychopy.org) and a custom wrapper to manage their sequential presentation and alignment with eye-tracking videos. The wrapper and stimuli are both available at https://github.com/ScottHarris17/Bassoon; (*Harris, 2022*). Aberrations in the projection were corrected by applying a manually fit spherical morph to all stimuli (Meshmapper, https://www.paulbourke.net). Blackout curtains surrounded the rig to minimize light contamination.

## Unidirectional OKR stimuli

Unidirectional sinusoidal gratings were presented in groups of six consecutive epochs. Each epoch consisted of 20 s of a static grating, followed by 60 s of a grating drifting either directly upward or directly downward, and an additional 20 s of a static grating. The six total epochs consisted of three upward and three downward epochs that were randomly interleaved. All gratings moved at 10°/s and had a spatial frequency of 0.15 cycles/°. The brightest part of the grating evoked $1.38 \times 10^3$ S-cone photoisomerizations/s and the darkest part of the grating evoked 25.9 S-cone photoisomerizations/s. M-cone photoisomerizations were 60% those of S-cone.

## Oscillating OKR stimuli

Oscillating sinusoidal gratings were presented in groups of three consecutive epochs. Each epoch consisted of 20 s of a static grating, followed by 120 s of oscillation, and an additional 20 s of a static grating. During the oscillation, the grating velocity was modulated sinusoidally up and down. The oscillation had an amplitude of 20°, a period of 15 s, and a phase shift of 0°. Eight oscillations were completed over the course of one epoch. All gratings had a spatial frequency of 0.15 cycles/°. The intensities of high-contrast oscillating gratings were equivalent to those used for unidirectional OKR stimuli and had fivefold greater Michelson contrast than low-contrast oscillating gratings (i.e., low-contrast gratings were '20% relative contrast'). High- and low-contrast gratings had the same mean luminance.

## Eye tracking

Prior to eye-tracking experiments, animals underwent stereotaxic surgery for implantation of a custom head-fixing apparatus. After surgery, animals were given 7 days to recover. Animals were then gradually habituated to the behavior rig by spending increasing amounts of time head-fixed on the rig for five consecutive days prior to the beginning of experiments.

Eye-tracking experiments were run over the course of up to 3 days per animal. Each animal spent no more than 30 min per day on the rig. During experiments, mice were head-fixed in the center of the hemisphere, which filled the entirety of their visual field. Eye movements were recorded using a GigE camera (Photonfocus), an infrared filter, and a hot mirror (Edmund Optics) that allowed the camera to be positioned outside of the animal's field of view. Infrared LEDs (880 nm) were mounted on the top and side of the camera to generate corneal reflections that marked the meridian and equator of the eye, respectively. StreamPix software (NorPix) was used to capture video of the eye and align it to stimuli via TTL signals.

After completion of the experiments, Deeplabcut (https://www.deeplabcut.org) was used to train a neural network to locate the pupil and corneal reflection on each video frame. The two-dimensional pupil location was then translated to angular eye position for every recording frame using the methods described by *Stahl et al., 2000*; *Stahl, 2004* and *Zoccolan et al., 2010*. In short, prior to experiments, we calibrated the eye tracking system for each animal by repeatedly swinging the camera ±6° in the horizontal plane and measuring the relative position of the pupil and corneal reflection. This process was repeated across five different luminances to fit a linear regression between pupil size and angular eye position. The meridian and equator of the eye were measured by turning on the top- and side-aligned infrared LEDs in sequence. During experiments, only the top LED was on. Angular eye position was then calculated as

$$\phi = arcsin\left(\frac{\Delta x}{\sqrt{R_{p0}^2 - \Delta y^2}}\right) \tag{1}$$

and

$$\theta = arcsin\left(\frac{\Delta y}{R_{p0}}\right) \tag{2}$$

where $\phi$ is the horizontal eye position, $\theta$ is the vertical eye position, $\Delta x$ is the horizontal distance measured between the eye center and the pupil, $\Delta y$ is the vertical distance measured between the eye center and the pupil, and $R_{p0}$ is the radius of rotation between the pupil and the eye's center,

which was computed empirically for each pupil diameter for each animal. For all stimuli, fast and slow nystagmus were separated on the basis of eye velocity and acceleration using custom MATLAB scripts. For all analyses in this report, we consider only $\theta$, the vertical eye position, and $\Delta\theta/\Delta t$, the vertical component of eye velocity.

By measuring the vertical eye velocity ($\Delta\theta/\Delta t$) in response to the static gratings (multiple contrasts) that occurred before the onset of all stimuli, we also computed a baseline median eye drift of 0.0787°/s (IQR: 4.1792°/s; p=0.015) in the ventral direction across animals. This drift may reflect either the error associated with the estimation of the eye's center during our calibration process or a natural biological eye drift associated with our rig. It is not possible to disambiguate between these possibilities. Since the eye position was near neutral for the time in which this drift was calculated, we baseline subtracted it from the eye position traces for all oscillating grating stimuli (for which the average eye position was also approximately neutral). For unidirectional gratings, we were unable to calculate an appropriate drift for baseline subtraction since the eye position was not often near neutral during these stimuli (*Figure 1—figure supplement 2*, *Figure 8—figure supplement 4*).

For oscillating OKR stimuli, saccades were removed from the eye trace post hoc. This was achieved by setting $\Delta\theta/\Delta t$ during saccades to its value immediately prior to the saccade onset, and then reintegrating $\Delta\theta/\Delta t$ to compute the position of the eye across time.

## Retrograde labeling

MTN-projecting retinal ganglion cells were labeled via stereotaxic injection of red fluorescent retrobeads (Lumafluor) into MTN. Prior to surgery, mice were anesthetized by IP injection of ketamine/ xylazine and 5% isoflurane inhalation. Once fully anesthetized, as assessed by absence of the pedal reflex, animals were transferred to a sterile-heated surface and the eyes were covered with a lubricating ointment. Then, 2% isoflurane was administered continuously to maintain anesthesia. Fur was removed prior to incision and lidocaine (<7 mg/kg) was injected locally under the scalp. After incision, animals' heads were leveled by aligning bregma and lambda in the horizontal plane. A burr hole was drilled at A/P: 0.00 mm, M/L: 0.85 mm from bregma. All injections were performed into the right MTN. A glass needle filled with retrobeads (diluted 1:3 in distilled water) and connected to a Hamilton syringe was lowered into the burr hole at an angle of 30° A/P to a depth of 5.36 mm below the surface of the brain. After 10 min, an injection of 400 nL was made at a rate of 5 nL/s. After injection, the needle was left in place for an additional 10 min before removal. The scalp was sutured and animals recovered in a heated cage. Analgesics (buprenorphine [0.05–0.1 mg/kg] and NSAIDs [5–10 mg/kg]) were delivered via subcutaneous injection immediately after animals awoke from anesthesia, again 12 hr later, and a third time the following morning. Animal health was monitored for 3 days after surgery and additional analgesics were administered as required. Labeling of retinal ganglion cells in the contralateral eye was typically observed as soon as 48 hr following surgery and did not increase or decrease with time.

## Empirical mosaic analysis

All mosaic analyses occurred ≥2 days after injection of the retrograde tracer into MTN. Retinas were dissected, fixed in 4% PFA for 20 min, and flat-mounted onto a microscope slide with a spacer. One widefield fluorescent image was taken of each retina. The location of each labeled cell in the image was determined using custom MATLAB scripts. Only retinas with near complete labeling (determined as greater than 500 identified RGCs) were included in analyses. The retina perimeter, optic nerve head, and dorsal-ventral axis were manually measured. These points were used to define a normalized polar coordinate system that allowed for the comparison of cell locations and densities across multiple retinas. Density recovery profiles were calculated using the methods described by *Rodieck, 1991*.

## Mosaic models

To model spatial distributions of single and multiple mosaics, we randomly generated mosaics in a model circular retina that had an equivalent radius to that of the average whole-mount retina used for empirical density recovery profile estimation. 'Cell bodies' were modeled as circles with radius 15 μm and scattered randomly across the model retina so long as the following conditions were met: (1) no two cell bodies can overlap in space (i.e., cells must form a 'monolayer'), and (2) adjacent cells that are members of the same mosaic must obey a (noisy) exclusion zone that is set by the mosaic coverage

factor (number of cells/retina area). Coverage factors were changed systematically such that the total number of cells across all mosaics – regardless of the number of mosaics being modeled – always approximated the number of retrogradely labeled cells per retina in our empirical data set. Density recovery profiles were computed as described for empirical data.

## Electrophysiology tissue preparation

All electrophysiology experiments occurred ≥2 days after injection of the retrograde tracer into MTN. Prior to electrophysiology experiments, mice were dark-adapted for ≥12 hr. Mice were then euthanized by cervical dislocation and the left eye was enucleated (contralateral to the right MTN injection). Retina dissections occurred in the dark using infrared converters and warmed bicarbonate-based Ames solution, equilibrated with 95% $O_2$/5% $CO_2$. Brains were simultaneously harvested and fixed in 4% PFA for imaging and to confirm that the retrograde tracer was properly injected into MTN. Retinas were whole-mounted, keeping track of orientation, and continuously perfused at 10 mL/min with freshly equilibrated Ames heated to 35°C throughout the course of experiments.

## Retinal location of recorded cells

At the beginning of electrophysiology experiments, the center and radius of the retina were measured: two-dimensional coordinates of eight standardized points around the perimeter of the retina were noted. The center of the retina was estimated by computing the median of the circumcenters of all unique triangles that could be formed from these eight points. The radius of the retina was estimated by finding the median radius of the circles that circumscribed these same triangles. In all cases, the convex hull of the whole-mount retina was well approximated by a circle and the retina center estimation was near the optic nerve head. The coordinates of each recorded cell were computed in reference to the retina center. Cell locations were combined across retinas by normalizing to the estimated radius in a polar coordinate system.

## Identification of MTN-projecting RGCs

An NIR light source (950 nm, Sutter) was used to visualize the tissue for the majority of the experiment. To identify retrogradely labeled ganglion cells, a green epifluorescent light was turned on briefly (~1–3 s) prior to recording from each cell. This light evoked a moderate number of spikes in oDSGCs. At least 1 min of darkness was provided between the offset of the epifluorescent light and the beginning of subsequent experiments. The epifluorescence exposure likely contributed variance to our dataset by differentially modulating adaptation states along the dorsal/ventral retinal axis as the absorption spectra of cone photoreceptors change. However, for all electrophysiology experiments, care was taken to record from comparable spatial distributions of Superior and Inferior oDSGCs such that reported asymmetries between cell types cannot be attributed to uneven proportions of Superior and Inferior oDSGCs recorded from dorsal and ventral retina. Further, repeated exposures to epifluorescence throughout an experiment had no effect on oDSGC responsivity. Moreover, asymmetries between Superior and Inferior oDSGC tuning curves were observed within each retinal quadrant and between pairs of Superior and Inferior oDSGCs (within 30 μm of each other) across the retina (*Figure 2—figure supplement 4*).

For a subset of cell-attached experiments, two-photon targeting was employed to validate and replicate our central findings. In these experiments, retrogradely labeled retinal ganglion cells were targeted on a two-photon microscope with peak emission at 860 nm and laser power of approximately 17–40 mW. Two-photon and epifluorescence targeting were never performed on the same retina. Data collected during two-photon targeting are presented *only* in *Figure 2—figure supplements 3 and 5*, *Figure 7—figure supplement 2*, *Figure 8E–G*, and *Figure 8—figure supplement 2*. The figure legends also clearly indicate experiments in which two-photon targeting was used. Unless otherwise stated, electrophysiology data came from experiments in which epifluorescence was used.

## Electrophysiology

Patch electrodes were pulled from borosilicate glass (Sutter) to 3–5 MOhm resistance using a Narishige puller. A MultiClamp 700B Amplifier (Axon Instruments) with acquisition rate of 10 kHz was used for all recordings. Cell-attached experiments were performed using electrodes filled with HEPES buffered

Ames. Voltage-clamp experiments were performed using fresh electrodes filled with cesium methane-sulfonate (*Care et al., 2019*). Current-clamp experiments were performed using fresh electrodes filled with potassium aspartate (*Care et al., 2020*). A subset of cells were recorded in both cell-attached and whole-cell configurations. In these cases, cell-attached recordings were performed first. Voltage-clamp and current-clamp recordings were never made from the same cell. Electrophysiology experiments were conducted using Symphony DAS (https://symphony-das.github.io/), and light stimuli were constructed and presented using Stage (https://stage-vss.github.io/).

### Light increment stimulus

Light increments were the first stimulus to be presented to each cell and were often additionally interleaved between other stimuli. Light increments were delivered for 1 s from darkness using an LED with peak emission at 405 nm. The increments had an intensity of $8.6 \times 10^4$ S-cone photoisomerizations/s. M-cone photoisomerizations were 79% of those of S-cone. The LED spot had diameter 500 µm or 300 µm (no significant difference was observed in the responses to either spot size), was centered on the cell body of each recorded cell, and was focused on the photoreceptor layer of the retina.

### Drifting bar stimulus

Drifting bars were presented from a DLP projector with peak emission at 405 nm (LightCrafter through EKB Technologies, same model as used for behavioral experiments). The native optics of the projector were replaced with neutral density filters and optics to focus stimuli on the photoreceptor layer of the retina via a condenser. The projector covered a rectangular area of 427 × 311 µm that was centered on the soma of each recorded cell. Drifting bars had a width of 3.2° and moved at 10°/s using a conversion factor of 31 µm/° (*Remtulla and Hallett, 1985*). Bar height was limited only by the area covered by the projector. High-contrast bars measured $2.4 \times 10^4$ S-cone photoisomerizations/s and were presented on top of a background of 124 S-cone photoisomerizations/s (for drifting bar experiments utilizing two-photon targeting [*Figure 2—figure supplements 3 and 5*, *Figure 7—figure supplement 2*] the background was $1.9 \times 10^3$ S-cone photoisomerizations/s). M-cone photoisomerizations were 74% of those of S-cone. See below for the specifications of low-contrast bars. For tuning curve estimation, bars moved in eight directions separated by 45°. The presented sequence of stimulus directions was randomized for each recording. Tuning curves were estimated by mean measurements taken over five repetitions per stimulus direction.

### Retinal ganglion cell classification

Retrogradely labeled retinal ganglion cells were classified as either Superior or Inferior oDSGCs if they had a direction selectivity index of greater than 0.05 and a preferred direction more than 30° away from the temporal-nasal axis, as calculated by spike outputs measured in the cell-attached configuration. The vast majority of recorded cells met these criteria, but those that did not were excluded from further analyses (*Figure 2—figure supplement 1*). All recorded cells in our data set were dominated by ON responses to a light increment (*Figure 3—figure supplement 2*).

### Current injections

Depolarizing or hyperpolarizing currents were continuously injected while measuring voltages across stimulus directions in the current-clamp configuration. The magnitude of current injections changed subtly from cell to cell depending on resting membrane potential, input resistance, and spike threshold. On average, depolarizing current injections increased the membrane potential by ~6 mV (to ~–48 mV), whereas hyperpolarizing current injections decreased the membrane potential by ~6 mV (to ~–60 mV). Depolarizing injections were always small enough such that the new resting membrane potential remained below spike threshold and each cell's baseline firing rate was 0 Hz. The order of depolarizing and hyperpolarizing injections was randomized across cells.

### Isolation of spikes and subthreshold voltages

From current-clamp recordings, the onset and offset of each action potential were determined using the first and second derivatives of the voltage trace and a fixed minimum refractory period. The subthreshold voltage was then linearly interpolated between action potential onsets and offsets.

## Subthreshold voltage tuning curves

The maximum voltage deflection from baseline was used to determine subthreshold membrane potential tuning curves. Values were averaged over five repetitions for each stimulus direction. Tuning curve metrics were calculated as for spikes.

## Electrophysiology at low contrast

For experiments using epifluorescence targeting (i.e., all data in *Figure 2*, *Figure 7*, and *Figure 7— figure supplement 1*), low-contrast drifting bars had an intensity of $0.5 \times 10^4$ S-cone photoisomerizations/s (approximately fivefold dimmer than high-contrast bars, or '20% relative contrast') from the same background of 124 S-cone photoisomerizations/s. For experiments using two-photon targeting (i.e., *Figure 7—figure supplement 2*), low-contrast bars had an intensity of $2.4 \times 10^4$ S-cone photoisomerizations/s and were presented on top of a background of $1.3 \times 10^4$ S-cone photoisomerizations/s. For both epifluorescence and two-photon targeting experiments, all other stimulus parameters were equivalent to what they were under high-contrast conditions. In a subset of cells, low-contrast bars failed to elicit spikes for every stimulus direction. In such cases, the cell's spike tuning curve area was set to 0, the area of its normalized tuning curve was set to 0, and its direction selectivity index was set to 1. We chose this convention because of the observation that, across cells, as the total number of spikes approached 0, the direction selectivity index approached 1 and the area of the normalized tuning curve approached 0. This pattern also fits the prediction made by our parallel conductance model (*Figure 7—figure supplement 1F*). Cells with no responses under low contrast were classified as Superior or Inferior on the basis of their responses to high-contrast stimuli.

## Immunohistochemistry

Individual ganglion cells were filled with either Lucifer yellow or biocytin during electrophysiology experiments. Retinas were subsequently fixed in 2% paraformaldehyde for 20 min at room temperature. The following protocol was used to enhance for the cell fills (anti-Lucifer yellow [Life Technologies A5750] and/or streptavidin-488 [Thermo Fisher S11223]), label synaptic puncta (anti-postsynaptic density [PSD-95, UC Davis NeuroMab 75-028], anti-Gephyrin [Synaptic Systems 147 111]), and stain for cholinergic starburst amacrine cells (anti-choline acetyltransferase [ChAT, Millipore AB144P]): blocking serum (1 day), primary antibody incubation (5 days), rinse 3× in PBS, secondary antibody (Jackson ImmunoResearch) incubation (1 day), rinse 3× in PBS. Retinas were mounted with a spacer in VECTASHIELD (Vector labs) and under a coverslip.

## Imaging

Individual oDSGCs with known direction selectivity and their associated synaptic puncta were imaged on a confocal microscope (Leica SP8) using a ×40 objective (NA 1.3) at a resolution of 0.102 × 0.102 × 0.3 μm.

## Image analysis

Confocal images were first median filtered in three dimensions (Fiji). Ganglion cell dendrites were reconstructed using the filament function in Imaris (Oxford Instruments). Convex polygons, dendritic branch numbers, and total dendritic length were obtained from the filaments. Excitatory PSD-95 and inhibitory gephyrin puncta were identified and quantified within the filament mask of the ganglion cell dendrites (ObjectFinder, https://lucadellasantina.github.io/ObjectFinder/; *Della Santina et al., 2013*).

## Electrical properties of oDSGCs

The resting membrane potential, spike threshold, and input resistance of oDSGCs were all measured during whole-cell current-clamp recordings. Resting membrane potential was taken as the initial membrane voltage immediately after establishing intracellular access. Spike threshold and input resistance were both calculated by injecting a slow ramp of current. Spike threshold was the average voltage at which a cell initiated its first action potential in response to the ramp. Input resistance was calculated from the average slope of the I-V response below spike threshold. Both metrics were averaged over at least five repetitions of the ramp stimulus per cell. The membrane capacitance was

measured during voltage-clamp recordings using the built-in capacitance calculator from the Multi-Clamp 700B Amplifier (Axon Instruments).

## Parallel conductance model

We implemented a parallel conductance model in MATLAB to build the model oDSGC (adapted from *Antoine et al., 2019*). Excitatory (*Gex*) and inhibitory (*Gin*) conductances were calculated at each time point for each direction of stimulus motion using Ohm's law:

$$G = I/\left(Vhold - Erev\right) \quad (3)$$

where $I$ is the mean current trace recorded in voltage-clamp across both Superior and Inferior cells, *Vhold* is the holding potential, and *Erev* is the reversal potential for either excitation or inhibition. A liquid junction potential of 5 mV was subtracted from *Vhold* and *Erev*. Because our model called for directionally untuned excitation, but the recorded excitatory conductances were slightly different for each direction of stimulus motion (likely due in part to space-clamp error), we used an identical excitatory conductance for all directions of stimulus motion that was equal to the maximum conductance at each time point across all recorded directions (space-clamp errors reduce empirically recorded excitatory currents in voltage-clamp mode). The excitatory conductance was then multiplied by a gain value to achieve a final time series for *Gex*. Next, we used the equation

$$C\frac{dV}{dt} = Gex\left(Eex - Vm\right) + Gin\left(Ein - Vm\right) + Grest\left(Erest - Vm\right) \quad (4)$$

to determine the membrane potential at each time point. $C$ is the median capacitance of Superior and Inferior oDSGCs as measured during whole-cell recordings. *Grest* is the reciprocal of the median input resistance, which we calculated by injecting a slow ramp of current in a subset of recorded cells and determining the average slope of the I-V response below spike threshold. *Erest* is the median resting membrane potential. *Vm* is calculated at each point in time by initializing it at *Erest*, and then determining each subsequent value using Euler's method and an integration time step of 1 ms. The peak change in this *Vm* value above *Erest* was used to construct the Vm tuning curves in the version of the model without a spiking component.

In the version of the model with a spiking component, we again solved for *Vm* at every time point using Euler's method. In this case, however, whenever *Vm* surpassed the threshold potential, a 'spike' was counted and a 3 ms pause corresponding to the spike time and refractory period was initiated, after which *Vm* was reset to *Erest* and the process continued. The threshold value was fixed such that the normalized area and direction selectivity index of the model oDSGC's spike tuning curve matched the median empirical values for these metrics (taken from cell-attached recordings) when the excitatory gain was set to 1.0. This resulted in a threshold value of –46.2 mV, which was 2.9 mV more negative than the median empirically recorded spike threshold.

Because resetting *Vm* to *Erest* after each spike also changed the driving forces for excitation and inhibition – and therefore possibly the shape of resulting spike tuning curves – we also simulated spike responses by assuming that the number of spikes produced to a given stimulus was linearly proportional to the amount of time that *Vm* (calculated without a spiking mechanism in place) spent above spike threshold (data not shown). Results from this model were not substantively different than those from the model in which a refractory period was included and *Vm* was reset to *Erest* after each spike.

This same model was used to test the contrast dependence of spike and Vm tuning curves. In this case, both the excitatory and each of the eight inhibitory conductance time series were multiplied by the gain value to calculate *Gex* and *Gin*, respectively.

## Behavioral predictions from drifting bar stimulus

Predictions for OKR gain were calculated on the basis of the difference in median firing rates between Superior and Inferior oDSGC populations to the drifting bar stimulus. The preferred and null directions of each cell in our dataset were computed in response to high-contrast drifting bars. Gain predictions were made for high- and low-contrast (20% relative) stimuli moving in the superior and inferior directions (i.e., four total conditions) by (1) resampling from distributions of oDSGC responses for that condition 10,000 times, (2) computing the difference (i.e., 'delta') between the median preferred and null direction responses on each iteration for the appropriate cell types, and (3) using the median of

these bootstrapped distributions of delta as an estimate of relative gain (*Figure 8—figure supplement 1*). The amplitudes of the predicted eye movements shown in *Figure 8C* reflect these predictions, while the sinusoidal trajectories are inferred from the stimulus motion.

Computing the preferred and null direction of each cell has the advantage of controlling for the fact that the observed preferred direction of individual oDSGCs can change based on retinotopic location when the retina is flat mounted. Assigning the preferred direction to a constant stimulus direction across cells (e.g., dorsal-to-ventral motion on the retina for Superior oDSGCs) did not change the behavioral predictions (data not shown).

## Behavioral predictions from oscillating grating stimulus

oDSGC responses were measured in the cell-attached configuration in response to the same oscillating grating stimulus used in behavioral experiments. Two-photon targeting was used for all oscillating grating electrophysiology experiments. Stimulus oscillations were sinusoidal and had a period of 15 s and an amplitude of 20°. The intensity of the grating was also sinusoidal across space, and had a spatial frequency of 0.15 cycles/°. For the high-contrast stimulus, the mean light intensity of the grating evoked $3.8 \times 10^3$ S-cone photoisomerizations/s, the peak intensity evoked $7.6 \times 10^3$ S-cone photoisomerizations/s, and the trough intensity evoked 124 S-cone photoisomerizations/s. For the low-contrast stimulus, the 20% relative (Michelson) contrast stimulus used in behavior experiments failed to evoke consistent spiking in oDSGCs. That 20% relative contrast gratings evoked OKR behavior but not oDSGC spikes can likely be explained by the fact that absolute contrasts and adaptation states were not matched between behavior and electrophysiology. However, assuming monotonic nonlinearities, it is only necessary to match the *direction* (e.g., higher to lower) of contrast change to predict the corresponding *direction* of behavioral change across contrasts. Thus, in electrophysiology experiments, we used a version of the low-contrast oscillating grating stimulus that was 50% relative contrast compared to the high-contrast stimulus described above. The mean light intensity of this stimulus evoked $3.8 \times 10^3$ S-cone photoisomerizations/s, the peak intensity evoked $5.7 \times 10^3$ S-cone photoisomerizations/s, and the trough intensity evoked $1.9 \times 10^3$ S-cone photoisomerizations/s. For all stimuli, M-cone photoisomerizations were 74% those of S-cones. The initial positional phase of the grating was randomized between cells.

Linear behavioral predictions were made from the spike responses of Superior and Inferior oDSGCs to these oscillating stimuli (*Figure 8—figure supplement 2*). The following procedure was repeated separately for responses to high- and low-contrast gratings: first, the median spike rate of Superior and Inferior oDSGCs was computed every 5 ms over the course of a single 15 s oscillation cycle (*Figure 8—figure supplement 2C and D*). A point-by-point subtraction was then performed (*Figure 8—figure supplement 2E and F*). The difference between the median spike rates of Superior and Inferior oDSGCs served as a linear prediction of eye velocity at each point in time. Therefore, predictions of eye position were computed across time by integrating these differences:

$$p\left(t\right) = \int \left(Sup. - Inf.\right) dt \tag{5}$$

where $p\left(t\right)$ is the vertical position of the eye at time $t$. The starting position of the eye was set to 0° (*Figure 8—figure supplement 2G and H*).

## Empirical nonlinearity

The relationship between linear predictions of OKR and measured eye velocities was estimated by finding the least-squares fit of a sigmoid function of the form

$$v\left(r\right) = v_{max} + \frac{\left(v_{max} - v_{min}\right)}{\left(1 + 10^{\left(r_{50} - r\right)m}\right)} \tag{6}$$

where $v_{min}$ is the minimum eye velocity, $v_{max}$ is the maximum eye velocity, $r_{50}$ is the difference in firing rate along the abscissa that corresponds to the inflection point, $m$ controls the slope, and $v\left(r\right)$ is the expected eye velocity for a given firing rate difference, $r$. For the drifting bar stimulus (*Figure 8D*), the linear predictions and behavioral eye velocities for superior and inferior stimuli at high- and low-contrast, along with a fifth point at the origin (0, 0), were used to fit the curve. For the oscillating

grating stimulus (*Figure 8G*), the high-contrast instantaneous linear predictions and the time-matched average eye velocities during high-contrast stimuli were used to fit the curve. Fit parameters for both curves are reported in the legend of *Figure 8*.

## Quantification and statistical analysis

### Statistics
All metrics reported in the text refer to population mean ± SEM unless otherwise specified. Nonparametric hypothesis tests were used to compute significance values wherever possible. Mann–Whitney U tests were used for instances in which the test type is not specified. Wilcoxon signed-rank and Fisher's exact tests were used where specified in the text and/or figure legends. All tests were two-sided. R values are Spearman's rank correlation coefficients. Lines of best fits are least-squares linear regressions. Significance markings are as follows: not significant (N.S.) for $p \geq 0.05$, $*p < 0.05$, $**p < 0.01$, $***p < 0.001$. All values can be found in the figures, figure legends, and 'Results' section. All statistical analyses were performed in MATLAB.

### Tuning curve area
Tuning curve area was calculated by dividing the area under the curve of the linear tuning curve by 360°. For clarity, this metric is also referred to as the 'linear tuning curve area'.

### Preferred direction
The preferred direction was calculated as the direction of the vector sum of spike responses to all eight stimulus directions of the drifting bar. Thus, the preferred direction was not necessarily equivalent to the single stimulus direction that evoked the largest response. The null direction was defined as 180° away from the preferred direction.

### Normalized tuning curves
Normalized tuning curves were calculated by first determining the cell's preferred direction (see above), and then dividing the response in all eight stimulus directions to the response in that preferred direction. For cases in which the preferred direction did not match a stimulus direction that was specifically probed, the preferred direction response was estimated by a linear interpolation of the two neighboring probed directions. The area of the normalized tuning curve (abbreviated as the 'normalized area') was calculated by dividing the area under the curve of the linear normalized tuning curve by 360°. The area of the normalized tuning curve is always greater than 0, but has no upper bound – though it tended to fall below 1. Larger values indicate a wider tuning curve. Perfectly circular tuning curves take a value of 1.

### Direction selectivity index
The direction selectivity index (DSI) was calculated as the magnitude of the vector sum divided by the scalar sum of responses to all eight stimulus directions. The direction selectivity index ranges between 0 and 1, with larger values indicating sharper tuning curves.

### Von Mises fit
Tuning curves were fit to the Von Mises function by minimizing the sum of squared residuals. The Von Mises function is a circular analog of the Gaussian curve defined as

$$f(x) = \frac{e^{\kappa cos(x-\mu)}}{2\pi I_0 (\kappa)} \tag{7}$$

where μ is the center of the curve, $1/\kappa$ controls the width of the curve, and $I_0$ is the modified Bessel function of the first kind, of order 0. A larger κ value indicates a sharper fit.

## Acknowledgements
We thank Connie Chen and Jeremiah John for technical assistance; Annika Balraj, Kevin Bender, David Copenhagen, Luca Della Santina, Daniel Denman, Tonatiuh Garcia Ruiz, Jonathan Horton, Jeanette

Hyer, David Kastner, Joo Yeun Lee, Yien Ming-Kuo, Satoru Miura, Yvonne Ou, Massimo Scanziani, Manuel Solino, and Alfred Yu for helpful discussions; David Berson, Guy Bouvier, and Fred Rieke for comments on the manuscript. This work was supported by NIH/NEI through F31 EY-033225 (SCH), R01 EY-029772 (FAD), R01 EY-030136 (FAD), and the Vision Core Grant P30 EY-002162 (UCSF), internally by the Moritz-Heyman Discovery Fellowship (SCH) and the Kavli Institute for Neuroscience (UCSF), and through foundation grants from the McKnight Scholar Award (FAD), Research to Prevent Blindness (Unrestricted Grant), and That All May See (UCSF).

## Additional information

### Funding

| Funder | Grant reference number | Author |
| --- | --- | --- |
| National Institutes of Health | F31 EY-033225 | Scott C Harris |
| National Institutes of Health | R01 EY-029772 | Felice A Dunn |
| National Institutes of Health | R01 EY-030136 | Felice A Dunn |
| Moritz-Heyman Discovery Fund | Student Fellowship | Scott C Harris |
| Kavli Institute for Neuroscience | Student Fellowship | Scott C Harris |
| McKnight Endowment Fund for Neuroscience | Scholar Award | Felice A Dunn |
| Research to Prevent Blindness | Unrestricted Grant | Felice A Dunn |

The funders had no role in study design, data collection and interpretation, or the decision to submit the work for publication.

### Author contributions

Scott C Harris, Conceptualization, Resources, Data curation, Software, Formal analysis, Funding acquisition, Validation, Investigation, Visualization, Methodology, Writing – original draft, Project administration, Writing – review and editing; Felice A Dunn, Conceptualization, Resources, Data curation, Formal analysis, Supervision, Funding acquisition, Investigation, Visualization, Methodology, Writing – original draft, Project administration, Writing – review and editing

### Author ORCIDs

Scott C Harris http://orcid.org/0000-0002-3567-4481
Felice A Dunn http://orcid.org/0000-0003-0784-0259

### Ethics

The study was performed in accordance with recommendations and protocols approved by the University of California, San Francisco Institutional Animal Care and Use Program (AN184353-02A). All surgery was done under anesthesia and all efforts were made to minimize suffering.

### Decision letter and Author response

Decision letter https://doi.org/10.7554/eLife.81780.sa1
Author response https://doi.org/10.7554/eLife.81780.sa2

## Additional files

### Supplementary files
• MDAR checklist

## Data availability

All data reported in this paper is publicly available at https://doi.org/10.7272/Q6RV0KZ3. All original code for visual stimulus generation and confocal image analysis has been deposited on GitHub and is publicly available. The information is listed in the key resources table.

The following dataset was generated:

| Author(s) | Year | Dataset title | Dataset URL | Database and Identifier |
|---|---|---|---|---|
| Harris SC, Dunn F | 2023 | Asymmetric retinal direction tuning predicts optokinetic eye movements across stimulus conditions | https://doi.org/10.7272/Q6RV0KZ3 | Dryad Digital Repository, 10.7272/Q6RV0KZ3 |

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
