## [Editor Report]

The optokinetic reflex (OKR) is a behavioral response that moves the eye so as to keep the image on the retina stabilized during the animal's movements. This important study traces the origins of that behavior down to cellular mechanisms of circuits in the retina. Specifically, it offers compelling evidence that the asymmetries and tuning properties of the OKR are shaped by the excitation/inhibition balance in retinal circuits and the regulation of spiking thresholds across different neuron types.

---

## [Decision Letter]

**Decision letter after peer review:**

Thank you for submitting your article "Asymmetric retinal direction tuning predicts optokinetic eye movements across stimulus conditions" for consideration by *eLife*. Your article has been reviewed by 3 peer reviewers, including Markus Meister as Reviewing Editor and Reviewer #1, and the evaluation has been overseen by Tirin Moore as the Senior Editor. The following individual involved in review of your submission has agreed to reveal their identity: Keisuke Yonehara (Reviewer #2).

General recommendations:

1 – There are too many figures. The main text figures alone contain 125 panels, not counting small insets. The reader is likely to miss the important results in a sea of less relevant information. The paper would be more readable with half the number of panels.

2 – Small effects: Many of the panels report tiny effect sizes, e.g. 1I, 1J, 1K etc. The differences may be statistically significant (small p-values), owing to the large number of cells examined, but they are small on an absolute scale, or relative to the variance in the cell population. Sometimes, effects of the same magnitude are declared non-significant in one condition and significant in another (e.g. 3K, 3L). The paper could be more convincing if it just focused on robust effects.

Major concerns:

3 – Relation of physiology to behavior: The OKR responses for uni-directional gratings (Figure 1C) and oscillating gratings (Figures1G-H) look strikingly different, with more reliable responses in both directions to the oscillating gratings. The physiology in this study is derived from drifting gratings but ultimately used to explain the behavior under oscillating gratings. To reconcile the two, it would be useful to see some oDSGC spiking responses to oscillating gratings used in the behavioral experiments. Do they reveal the same asymmetries between inferior and superior oDSGCs?

4 – Morphology and space clamp, Line 247-252: The authors found that the spike-to-EPSC ratio (Figure S4P) and input resistance were lower in superior oDSGCs, and claimed that superior oDSGCs are less excitable. As the authors discuss, the morphological difference (size of the dendritic field) between superior and inferior oDSGC types can cause space clamp problems and an apparent difference in EPSC. To validate the intrinsic excitability, the authors should measure the membrane excitability or gain by recording of membrane potentials while applying electrical stimulation, ranging from subthreshold to saturation level.

5 – Effects of current injection, Figure 5: The authors describe that injected currents that induce ~6 mV changes did not cause baseline firing. However, there may be subthreshold biophysical effects of this depolarization, such as inactivation of LVA ca^2+^ channels, known to exist in certain RGCs (Henderson and Miller, Vis Neurosci, 2003; Huang and Li, J Neurosci Res, 2006). Steady hyperpolarization may de-inactivate Na^+^ channels (Hodgkin and Huxley, J. Physiol. 1952; Armstrong and Bezanilla, J. Gen. Physiol. 1974). The authors should consider contribution of such sub-threshold mechanisms to visual excitability.

6 – Interpretation of current injection, Figure 5: The steady current injection experiments in Figure 5 are informative for exploring spike thresholding. However, this manipulation is different from the actual difference of EPSCs between superior and inferior oDSGCs, which are only different during light-evoked responses but not during the baseline period. To pursue this one could examine the correlation between EPSC strength and spiking tuning width/direction selectivity across all oDSGCs without current injection. According to the spike thresholding mechanism, oDSGCs receiving stronger EPSCs would show broader tuning and decreased direction selectivity.

7 – Contrast nonlinearities, Figures 3, 5, and 7: The authors examine the directional tuning of spikes and Vm in different contrast conditions. The differences between spikes and Vm indicate the nonlinearity in thresholding contributes to the tuning changes. However, it is not clear if the contrast modulates synaptic inputs. Specifically, if bipolar cells or starburst cells have a contrast-dependent nonlinearity in the neurotransmitter release, the tuning can be affected by the contrast changes. The authors should examine or discuss the possible modulation in synaptic inputs by stimulus contrast.

8 – Contribution of inhibition: In Line 194, the authors report that "inhibitory inputs failed to account for the differences between superior and inferior oDSGCs". However, in Figure 3, it appears that compared to inferior oDSGCs, IPSCs of superior oDSGCs are more narrowly tuned (Figures3C and 3D), while EPSCs are more broadly tuned (with two local peaks just like the spiking activity in Figures3H-I). The more narrowly tuned inhibition could contribute to the broader tuning of superior oDSGCs. Similarly, the difference in excitation between the two cell types appears to go beyond a simple "untuned" excitation because their excitatory tuning curves show bimodal vs unimodal distributions that match their spike tuning curves. These points are worth more discussion.

9 – Relative importance of gain and tuning width: Much of the manuscript elaborates on two superior-inferior differences: the absolute firing rate of the response and the width of the tuning curve. The final Figure 8 relating physiology to behavior seems to be based solely on firing rate differences. Is there a functional consequence to the difference in tuning width between the two cell types as it relates to the behavior? If not, the manuscript might be simplified by focusing on the gain.

---

## [Author Response]

General recommendations:1 – There are too many figures. The main text figures alone contain 125 panels, not counting small insets. The reader is likely to miss the important results in a sea of less relevant information. The paper would be more readable with half the number of panels.

We thank the reviewers for the suggestion to streamline the results. To do so we have removed panels from every main figure. The revised manuscript contains 75 panels in the main figures.

2 – Small effects: Many of the panels report tiny effect sizes, e.g. 1I, 1J, 1K etc. The differences may be statistically significant (small p-values), owing to the large number of cells examined, but they are small on an absolute scale, or relative to the variance in the cell population. Sometimes, effects of the same magnitude are declared non-significant in one condition and significant in another (e.g. 3K, 3L). The paper could be more convincing if it just focused on robust effects.

Consistent with the comment above, we have removed several panels from the main figures to focus on the most robust effects. We have also addressed some sources of variance within cell types, for example by examining metrics across retinal topography (Figure 2 - figure supplement 4-5). In some cases, different metrics and axis scales may make effect sizes appear visually more similar than they are (.g. Figure 3I, J [previously 3K, L]). Where possible, we have matched the scale of axes across conditions (e.g. Figure 3C, G) so that the effect size is illustrated more clearly.

Major concerns:3 – Relation of physiology to behavior: The OKR responses for uni-directional gratings (Figure 1C) and oscillating gratings (Figures1G-H) look strikingly different, with more reliable responses in both directions to the oscillating gratings. The physiology in this study is derived from drifting gratings but ultimately used to explain the behavior under oscillating gratings. To reconcile the two, it would be useful to see some oDSGC spiking responses to oscillating gratings used in the behavioral experiments. Do they reveal the same asymmetries between inferior and superior oDSGCs?

We thank the reviewers for the suggestion to strengthen our results by measuring the responses of oDSGCs to oscillating gratings. The revised manuscript includes cell-attached spike recordings from Superior and Inferior oDSGCs in response to oscillating gratings (Figure 8, Figure 8 - figure supplement 2). The data show that Superior oDSGCs respond preferentially to the upward movement of the grating, whereas Inferior oDSGCs respond preferentially to the downward movement (Figure 8E, Figure 8 - figure supplement 2C-D). Moreover their responses are asymmetric, with larger responses in Superior compared to Inferior oDSGCs. These results qualitatively match our findings from the drifting bar stimulus. We have also used these data to make an additional linear prediction of OKR behavior by subtracting the instantaneous firing rates of Superior and Inferior oDSGCs over the course of a single oscillation cycle (Figure 8F, Figure 8 - figure supplement 2E-H). These linear predictions closely match those made from responses to drifting bars, as well as OKR measured from behaving animals. This analysis is featured in the revised version of Figure 8.

4 – Morphology and space clamp, Line 247-252: The authors found that the spike-to-EPSC ratio (Figure S4P) and input resistance were lower in superior oDSGCs, and claimed that superior oDSGCs are less excitable. As the authors discuss, the morphological difference (size of the dendritic field) between superior and inferior oDSGC types can cause space clamp problems and an apparent difference in EPSC. To validate the intrinsic excitability, the authors should measure the membrane excitability or gain by recording of membrane potentials while applying electrical stimulation, ranging from subthreshold to saturation level.

As recommended, we have injected current into individual oDSGCs and recorded their change in membrane potential. Author response image 1 Panel A shows the input resistances ( V/I, or slope of the change in membrane potential) for each oDSGC type in response to a linear current injection ramp from 0 to 50 pA (the membrane potential remained below the spike threshold). Overall, Superior oDSGCs have a lower input resistance than Inferior oDSGCs, indicating that they are indeed less excitable (p = 0.002 by rank sum test). These data are reported in the manuscript (p8, ¶4, lines 240-253) and shown in Figure 4 - figure supplement 1.

To further validate this finding, we injected square current steps into Superior and Inferior oDSGCs, as suggested by the reviewers. Author response image 1 Panel B shows the resulting mean ± SEM impulse response functions for each oDSGC type. Superior oDSGCs have lower voltage-to-current ratios, indicating that they are less excitable.

In Author response image 1 panel C, we have calculated input resistances separately for the ramp and step experiments. These metrics were tightly correlated (Spearman R=0.87, p<0.001, n=11 oDSGCs), indicating that both manipulations probe the intrinsic excitability of oDSGCs similarly well. For this reason, and because of the comparatively low sample size in the current step experiment, we have elected not to include the current step data in the manuscript.

Finally, while space clamp problems may account for differences in EPSC direction tuning between Superior and Inferior oDSGCs, they are unlikely to explain the greater amount of excitatory input to Superior oDSGCs as shown in Figure 3G. All else equal, space clamp errors are expected to increase with the dendritic field size of a neuron(Spuston et al, J. Neurophysiol, 1993). Our data show that Superior oDSGCs have larger dendritic fields than Inferior oDSGCs (Figure 4B), and therefore likely suffer from greater space clamp errors. However, space clamp errors *decrease* EPSC magnitudes recorded at the soma. Therefore, space clamp errors are unlikely to explain the observation that EPSCs are greater in Superior oDSGCs than in Inferior oDSGCs, and instead suggest that the difference in EPSC magnitude between Superior and Inferior oDSGCs could be even larger than we measured.

**Author response image 1. sa2fig1:** Superior oDSGCs are less excitable than Inferior oDSGCs. (A) Input resistance in Superior (magenta) and Inferior (gray) oDSGCs calculated from current injection ramps in which current increases linearly from 0 to 50 pA over 0.25 seconds. (B) Steady state current-voltage relationship for Superior (magenta) and Inferior (gray) oDSGCs in response to current steps from -200 to 200 pA lasting 0.5 seconds each. (C) Input resistances calculated from current ramps and steps were positively correlated. For panel (C), R and p values are Spearman’s rank correlation coefficient and associated 2-sided p-value. Dashed line is least squares linear regression.

Spruston, N.., Jaffe, D. B., Williams, S. H., and Johnston, D (1993). Voltage-and space-clamp errors associated with the measurement of electrotonically remote synaptic events. Journal of neurophysiology, 70(2), 781-802.

5 – Effects of current injection, Figure 5: The authors describe that injected currents that induce ~6 mV changes did not cause baseline firing. However, there may be subthreshold biophysical effects of this depolarization, such as inactivation of LVA ca^2+^ channels, known to exist in certain RGCs (Henderson and Miller, Vis Neurosci, 2003; Huang and Li, J Neurosci Res, 2006). Steady hyperpolarization may de-inactivate Na^+^ channels (Hodgkin and Huxley, J. Physiol. 1952; Armstrong and Bezanilla, J. Gen. Physiol. 1974). The authors should consider contribution of such sub-threshold mechanisms to visual excitability.

We thank the reviewers for pointing out these subthreshold effects of current injection. To determine the effects of currents on the intrinsic properties of oDSGCs, we measured the peak rate of voltage change (dV/dt) during spikes while injecting either depolarizing or hyperpolarizing current. We found a slight increase in this rate of change in the hyperpolarizing condition, consistent with an increase in NaV channel availability. In agreement, the spike threshold potential was more negative during hyperpolarizing injections than during depolarizing injections. These data are shown in Figure 5 - figure supplement 2. We emphasize that these observations do not detract from the thresholding mechanism that is revealed by current injection experiments (Figure 5). The reason is that changes to NaV channel availability caused by current injection increased the excitability of neurons comparatively more during the *hyperpolarizing* condition. However, our experimental data show that oDSGCs were more excitable (i.e., produced more spikes and had broader tuning curves) during the *depolarizing* condition. This discrepancy is explained by the observation that, despite the change in NaV availability, all recorded cells’ resting membrane potential sat further away from their spike threshold during the hyperpolarizing condition compared to the depolarizing condition (Figure 5 - figure supplement 2D). Thus, changes in NaV availability are masked, and canceled to an extent, by the direct influence of current injections on the resting membrane potential.

To our knowledge, T-type calcium channel expression has not yet been reported in mouse oDSGCs. Previous reports of low voltage-activated calcium channels in RGCs of other species have indicated that activation of T-type channels may change by ~10-30% within the range that our current injections span (appx. -48 to -65 mV, Huang and Li, J Neurosci Res, 2006). It is difficult to speculate about the effects that such changes may have on our measurements, but we suggest that the most parsimonious explanation of the data shown in Figure 5 is given by the thresholding mechanism. Nonetheless, we have included a brief discussion of the impact of current injections on intrinsic properties of oDSGCs in the legend of Figure 5 —figure supplement 2.

6 – Interpretation of current injection, Figure 5: The steady current injection experiments in Figure 5 are informative for exploring spike thresholding. However, this manipulation is different from the actual difference of EPSCs between superior and inferior oDSGCs, which are only different during light-evoked responses but not during the baseline period. To pursue this one could examine the correlation between EPSC strength and spiking tuning width/direction selectivity across all oDSGCs without current injection. According to the spike thresholding mechanism, oDSGCs receiving stronger EPSCs would show broader tuning and decreased direction selectivity.

We thank the reviewers for this observation and agree that the current injection experiments do not replicate the stimulus dependence of the difference in EPSC magnitudes. We further agree that comparing EPSC magnitude with the spike tuning curve width should also reveal the effects of thresholding. In Author response image 2, panels A and B plot the EPSC tuning curve area against the DSI (A) and normalized area (B) of the spike tuning curve. The dashed lines show least squares linear regressions. The reviewers’ predictions are correct: for both metrics, tuning curves tend to be wider (i.e., lower DSI and larger normalized area) when EPSCs are greater. These trends are not significant in the existing dataset, likely because there is sizeable variance in other metrics that contribute to spike tuning curve shape (e.g., IPSC magnitude and tuning, threshold potential, etc.). However, none of these other contributors to tuning curve shape can explain the *differences between* the spike tuning curves of Superior and Inferior oDSGCs, as they are either similar across cell types (Figure 3C, J, Figure 3 - figure supplement 1A; Figure 4 - figure supplement 1C-D), or promote sharper, rather than broader, spike tuning curves in Superior oDSGCs (Figure 3D). Thus, the data demonstrate that EPSC magnitude is the dominant contributor to differences between Superior and Inferior oDSGC tuning (see response to point 8 below for a discussion of differences in EPSC and IPSC tuning between oDSGC types).

An additional way of addressing the reviewers’ predictions is to examine the spike tuning curve width as a function of the peak EPSP magnitude from current clamp recordings in which no current was injected. This relationship is shown in Author response image 2 panels C and D for the EPSP magnitude in each cell’s null direction. Inhibition tunes EPSPs, and we chose to examine the null direction because the relatively small EPSP magnitude in this direction means that this is the response for which thresholding should have its greatest influence on the shape of the spike tuning curve. EPSP magnitude is more strongly correlated with both spike DSI (C) and normalized area (D) than is EPSC magnitude (A, B) because the variance in a multitude of contributors to direction selectivity, including inhibition and cell-intrinsic properties, is already accounted for in the EPSP magnitude. Regardless, as suggested in panels A and B, larger EPSPs yield wider spike tuning curves. Together, these data agree with our previous analyses indicating that EPSC magnitude and thresholding are important contributors to oDSGC tuning curve shape. We have decided to not include the plots shown in Author response image 2 the revised manuscript because the data is limited and the results do not change our main points about thresholding.

**Author response image 2. sa2fig2:** Strength of excitatory input and spike tuning curve properties. (A-B) Relationships between excitatory postsynaptic current (EPSC) tuning curve area and (A) direction selectivity index or (B) normalized area of the spike tuning curve. Data taken from cells in which both voltage-clamp and cell-attached recordings were made. (C-D) Relationships between the peak subthreshold membrane potential in the cell’s null direction and (C) the direction selectivity or (D) normalized area of the spike tuning curve for cells recorded in the current-clamp configuration with no current injection. For all panels, R and p values are Spearman’s rank correlation coefficient and associated 2-sided p-value. Dashed lines are least squares linear regressions.

7 – Contrast nonlinearities, Figures 3, 5, and 7: The authors examine the directional tuning of spikes and Vm in different contrast conditions. The differences between spikes and Vm indicate the nonlinearity in thresholding contributes to the tuning changes. However, it is not clear if the contrast modulates synaptic inputs. Specifically, if bipolar cells or starburst cells have a contrast-dependent nonlinearity in the neurotransmitter release, the tuning can be affected by the contrast changes. The authors should examine or discuss the possible modulation in synaptic inputs by stimulus contrast.

We thank the reviewers for the opportunity to clarify this point. We agree that we have not directly measured whether contrast modulates the tuning of excitatory or inhibitory synaptic inputs to oDSGCs. However, if bipolar cell or starburst amacrine cell direction tuning changed across contrasts, we would expect to see a corresponding change to the shape of the membrane potential tuning curve. Instead, our data show that Vm tuning curves do not change shape across contrasts (Figure 7E, Figure 7 - figure supplement 1B). Thus, it is unlikely that the tuning of excitatory or inhibitory inputs changes across contrasts, though we cannot rule out the possibility that they do so in a way that precisely counteract each other. In further support of the idea that contrast does not modulate the tuning curves of synaptic inputs to oDSGCs, we show that E/I in the preferred and null directions is contrast invariant (Figure 7 - figure supplement 1D-E). In addition, oDSGCs share many of their presynaptic partners with ON-OFF DSGCs

(ooDSGCs), including starburst amacrine cells and bipolar cell types. If contrast modulates the tuning of these presynaptic neurons, this effect would likely produce contrast sensitivity in ooDSGC tuning curves. However, multiple groups have demonstrated that direction selectivity in ooDSGCs is contrast invariant (Poleg-Polsky and Diamond, J Neurosci 2016; Sethuramanujam et al., Neuron 2016; Sethuramanujam et al., Neuron 2017). For these reasons, we find it unlikely that presynaptic neurons change their direction tuning properties across contrasts.

More directly, however, even if synaptic inputs to oDSGCs are contrast-sensitive, this likely cannot account for the contrast-sensitivity of oDSGC spiking since the Vm tuning curves are contrast-invariant (Figure 7E, Figure 7 - figure supplement 1B). Thus, spike thresholding remains the only plausible mechanism to explain this phenomenon.

Poleg-Polsky, A., and Diamond, J.S. (2016). Retinal Circuitry Balances Contrast Tuning of Excitation and Inhibition to Enable Reliable Computation of Direction Selectivity. J. Neurosci. 36 , 5861–5876. 10.1523/JNEUROSCI.4013-15.2016.

Sethuramanujam, S., McLaughlin, A.J., deRosenroll, G., Hoggarth, A., Schwab, D.J., and Awatramani, G.B. (2016). A Central Role for Mixed Acetylcholine/GABA Transmission in Direction Coding in the Retina. Neuron 90 , 1243–1256.

10.1016/j.neuron.2016.04.041.

Sethuramanujam, S., Yao, X., deRosenroll, G., Briggman, K.L., Field, G.D., and Awatramani, G.B. (2017). “Silent” NMDA Synapses Enhance Motion Sensitivity in a Mature Retinal Circuit. Neuron 96 , 1099-1111.e3. 10.1016/j.neuron.2017.09.058.

8 – Contribution of inhibition: In Line 194, the authors report that "inhibitory inputs failed to account for the differences between superior and inferior oDSGCs". However, in Figure 3, it appears that compared to inferior oDSGCs, IPSCs of superior oDSGCs are more narrowly tuned (Figures3C and 3D), while EPSCs are more broadly tuned (with two local peaks just like the spiking activity in Figures3H-I). The more narrowly tuned inhibition could contribute to the broader tuning of superior oDSGCs. Similarly, the difference in excitation between the two cell types appears to go beyond a simple "untuned" excitation because their excitatory tuning curves show bimodal vs unimodal distributions that match their spike tuning curves. These points are worth more discussion.

We thank the reviewers for the suggestion to clarify the contributions of excitatory and inhibitory inputs on tuning curve shape. The intuition that more narrowly tuned inhibition can contribute to broader spike tuning is correct in some regimes. However, in other regimes, more narrowly tuned inhibition can also contribute to *sharper* spike tuning. The relationship between the direction selectivity index (DSI) of inhibition and the DSI of spikes is highly nuanced and varies according to the particular shape of the inhibitory tuning curve. Rather than attempt to explore this relationship exhaustively, we have compiled specific analyses that indicate that sharper inhibitory tuning in Superior oDSGCs is unlikely to contribute to their broader spike tuning curves.

Panel A in Author response image 3 shows that there is no significant correlation between inhibitory DSI and spike DSI in our dataset. This result may indicate that inhibition does not explain differences in the spike tuning curves of Superior and Inferior oDSGCs. However, the lack of correlation may also be attributable to the multitude of other factors that contribute to an oDSGC’s spike tuning ( e.g., magnitude of inhibition, magnitude and tuning of excitation, threshold potential, input resistance, etc.), each of which adds noise to this relationship. Thus, additional analyses are needed to parse the relationship between inhibitory tuning and spike tuning in oDSGCs:

In panels B and C we use our parallel conductance model to isolate the contribution of inhibition to oDSGC direction tuning. Panel B shows the DSI of a model oDSGC’s subthreshold membrane potential (Vm) and spikes as the DSI of the inhibitory conductance is progressively increased along the x-axis (excitation is held constant in this version of the model). The model predicts that, all else equal, narrower inhibitory tuning in oDSGCs yields narrower Vm and spike tuning curves. Panel C shows the output of two separate instantiations of the parallel conductance model for which inhibitory conductances were derived either only from Superior oDSGCs (Figure 3C, magenta line), or only from Inferior oDSGCs (Figure 3C, gray line). The model predicts that, all else equal, inhibitory conductances in Superior oDSGCs yield narrower Vm and spike tuning curves than those in Inferior oDSGCs. These modeling results indicate that inhibitory tuning is unlikely to account for the empirical finding that Superior oDSGCs have broader spike tuning curves.

Panel D shows that empirically measured Vm tuning curves (from current-clamp recordings) were more sharply tuned in Superior oDSGCs. Thus, as predicted by the model, narrower inhibitory inputs to Superior oDSGCs result in *narrower,* not wider, Vm tuning curves. In contrast, the finding that Superior oDSGCs have empirically broader spike tuning curves than Inferior oDSGCs can be accounted for by differences in excitation gain: excitation is greater in Superior oDSGCs (Figure 3G), resulting in larger Vm tuning curves (Author response image 3 panel E), and broader spike tuning curves via, among other mechanisms, thresholding. The data in panels D and E are replotted from Figure 5K-J and collapsed across conditions; the same statistics are marked by the arrowheads in those figures.

Collectively, these analyses all indicate that the narrower inhibitory tuning in Superior oDSGCs is unlikely to account for their broader spike tuning curves. The oDSGCs in our dataset likely reside in a particular regime of the inhibition DSI-spike DSI relationship where the two metrics are positively correlated.

With regard to the tuning of excitation, we agree with the reviewers that both the broader spike tuning of Superior oDSGCs and their bimodal tuning curves could be associated with the tuning properties of excitatory inputs. However, we also point out that bimodality of the spike tuning curve was observed in both Superior and Inferior oDSGCs following injection of directionally untuned depolarizing currents (Figure 5C-D). Thus, we are cautious in our interpretation, as it seems possible that this characteristic is also associated with excitation gain.

To address these points, we have indicated in the Discussion the likely contribution of narrower inhibitory tuning in Superior oDSGCs, though we have decided that the analysis in Author response image 3 are too involved to be included in the current manuscript and deserve further exploration in additional studies. We have also added an examination of the potential contributions of tuned excitatory inputs. The additions to the Discussion can be found on p18-19, ¶4, lines 586-615.

**Author response image 3. sa2fig3:** (A) Comparison of the direction selectivity index (DSI) computed for inhibitory inputs (from voltage-clamp recordings) and for spikes (cell-attached recordings) for cells in which both metrics were recorded. There is no significant relationship, indicating that inhibition is a poor predictor of spike tuning, but this may be caused by noise contributed by other circuit and cell-intrinsic processes. R and p values are Spearman’s rank correlation coefficient and associated 2-sided significance values, respectively. The dashed line is a least squares linear regression. (B) Using our parallel conductance model, the subthreshold membrane potential (Vm) and spike DSI were computed as a function of the DSI of inhibition. For oDSGCs, narrower inhibitory tuning curves predict narrower Vm and spike tuning curves. (C) Output of two additional iterations of the parallel conductance model in which inhibitory conductances were taken only from data recorded from Superior oDSGCs (magenta bars) or only from data recorded from Inferior oDSGCs (gray bars). All other parameters of the model (including excitation, input resistance, threshold potential, and resting membrane potential) were held constant across conditions. All else equal, the model predicts that the inhibitory inputs to Superior oDSGCs predict sharper Vm and spike tuning curves than the inhibitory inputs to Inferior oDSGCs do. (D) Direction selectivity indices of Vm tuning curves, recorded empirically in current-clamp mode. Superior oDSGCs have sharper Vm tuning curves than Inferior oDSGCs. This result matches the prediction of the model from (C), and indicates that sharper inhibitory tuning in Superior oDSGCs does not predict their broader spike tuning curves. These data are also shown in Figure 5K. (E) Area of linear Vm tuning curves recorded empirically in current-clamp mode. Superior oDSGCs have larger Vm tuning curves. This can be accounted for by their larger excitatory inputs (Figure 3G) and explains their broader spike tuning curves. These data are also shown in Figure 5J.

9 – Relative importance of gain and tuning width: Much of the manuscript elaborates on two superior-inferior differences: the absolute firing rate of the response and the width of the tuning curve. The final Figure 8 relating physiology to behavior seems to be based solely on firing rate differences. Is there a functional consequence to the difference in tuning width between the two cell types as it relates to the behavior? If not, the manuscript might be simplified by focusing on the gain.

Figure 8 relies on the difference in magnitude between the preferred and null direction responses of opposing cell types. The reviewers are correct in that this is largely related to the gain difference between cell types; however it is also associated with how sharply the cells are tuned (e.g., consider a common alternative definition of direction selectivity: (PD-ND)/(PD+ND)). Additionally, we suggest that differences in tuning curve width between Superior and Inferior oDSGCs will also have functional consequences for oblique OKR.